# Spatiotemporal proteomic profiling of the pro-inflammatory response to lipopolysaccharide in the THP-1 human leukaemia cell line

Claire M. Mulvey [1,5,8], Lisa M. Breckels [1,8], Oliver M. Crook [1,2], David J. Sanders [3], Andre L. R. Ribeiro[3], Aikaterini Geladaki [1], Andy Christoforou[4], Nina Kočevar Britovšek [1,6], Tracey Hurrell[1], Michael J. Deery[1], Laurent Gatto [1,7], Andrew M. Smith [3✉] & Kathryn S. Lilley [1✉]

Protein localisation and translocation between intracellular compartments underlie almost all physiological processes. The hyperLOPIT proteomics platform combines mass spectrometry with state-of-the-art machine learning to map the subcellular location of thousands of proteins simultaneously. We combine global proteome analysis with hyperLOPIT in a fully Bayesian framework to elucidate spatiotemporal proteomic changes during a lipopolysaccharide (LPS)-induced inflammatory response. We report a highly dynamic proteome in terms of both protein abundance and subcellular localisation, with alterations in the interferon response, endo-lysosomal system, plasma membrane reorganisation and cell migration. Proteins not previously associated with an LPS response were found to relocalise upon stimulation, the functional consequences of which are still unclear. By quantifying proteome-wide uncertainty through Bayesian modelling, a necessary role for protein relocalisation and the importance of taking a holistic overview of the LPS-driven immune response has been revealed. The data are showcased as an interactive application freely available for the scientific community.

[1] Cambridge Centre for Proteomics, Cambridge Systems Biology Centre and Department of Biochemistry, University of Cambridge, Cambridge CB2 1QR, UK. [2] MRC Biostatistics Unit, Cambridge Institute for Public Health, Forvie Site, Robinson Way, Cambridge CB2 0SR, UK. [3] Department of Microbial Diseases, Eastman Dental Institute, University College London, Royal Free Campus, Rowland Hill Street, London NW3 2PF, UK. [4] Bristol Myers Squibb, 10300 Campus Point Drive, San Diego, CA, USA. [5] Present address: Cancer Research UK Cambridge Institute, University of Cambridge, Li Ka Shing Centre, Cambridge CB2 0RE, UK. [6] Present address: Lek d.d., Kolodvorska 27, Mengeš 1234, Slovenia. [7] Present address: de Duve Institute, UCLouvain, Avenue Hippocrate 75, Brussels 1200, Belgium. [8] These authors contributed equally: Claire M. Mulvey, Lisa M. Breckels. ✉email: andrew.m.smith@ucl.ac.uk; k.s.lilley@bioc.cam.ac.uk

The compartmentalisation of proteins within membrane-bound, subcellular structures called organelles is a defining feature of eukaryotic cells. The function of a protein can often be inferred from its subcellular location and its availability to interact with other biomolecules within its microenvironment. A protein may exert multifunctional, context-specific roles if it resides in different locations within a single cell, a process known as "moonlighting"[1]. Dysfunctional protein localisation or the disruption of dynamic shuttling events between subcellular niches can underlie various pathophysiological processes and contribute to disease progression[2]. Protein trafficking and translocation play a major role in the innate immune response of mammalian cells[3], and although dysregulation of these events can contribute to an infection and the development of inflammatory diseases, the drivers of these processes remain incompletely characterised.

Lipopolysaccharides (LPS) derived from the outer membrane of Gram-negative bacteria are some of the most potent activators of the immune system. Toll-like receptor 4 (TLR4), the mammalian receptor for LPS, plays a beneficial role in controlling bacterial infections but is also a main driver of sepsis. It has recently been estimated that there are 49 million cases of sepsis per year globally, with 11 million sepsis-related deaths[4], and sepsis is the most common cause of death in people who have been hospitalised[5]. In addition, Alzheimer's disease, alcoholic liver disease, cardiovascular disease and type 2 diabetes have all been associated with systemic LPS exposure and subsequent inflammatory response[6,7]. LPS induces a distinct pro-inflammatory response resulting in activation of TLR4 signalling via MyD88-dependent and independent (TRIF-dependent) pathways, with an elevation in pro-inflammatory cytokines and Type 1 interferons (hereafter referred to as IFNs), respectively[8].

Previous studies have successfully used proteomics approaches to either explore aspects of the innate immune response to LPS in macrophages[9,10] or to characterise individual subcellular regions of interest following LPS-stimulation[11,12]. Despite these efforts, the cell-wide mechanism underlying the LPS-driven response is still not fully understood. Although assessing alterations in gene and protein expression is certainly important, the dynamic processes of protein trafficking and translocation may also underlie many abnormal physiological processes. We, therefore, decided to investigate subcellular protein translocation during the response to LPS stimulation, using the human monocytic leukaemia cell line THP-1, which is widely used in the immunology field and has proven particularly difficult to transfect and image by microscopy, due in part to its non-adherent morphology while in an undifferentiated, monocytic state.

Non-imaging mass spectrometry (MS)-based spatial proteomics methods provide an alternative yet orthogonal approach to microscopy for the investigation of protein subcellular localisation[13,14]. MS has provided a means of cataloguing the components of purified organelles[15,16], however, this approach may be prone to false-positive assignments due to difficulties associated with purifying organelles to homogeneity which have similar physicochemical properties. Recent advances in the field of spatial proteomics have taken a holistic approach to characterise and quantifying protein subcellular localisation. These methods provide a global overview of a cell population by accurately pinpointing thousands of proteins to distinct subcellular niches, enabling molecular annotation of whole organelles without the need for organelle purification[17–19]. One of the forerunners in this field is LOPIT—Localisation of Organelle Proteins by Isotope Tagging[20], and its more recent iterations, hyperLOPIT (hyperplexed LOPIT)[21–27] and LOPIT-DC (LOPIT after Differential ultraCentrifugation)[24,28]. The hyperLOPIT method combines extensive biochemical cell fractionation with multiplexed high-resolution MS-based proteomics for the simultaneous analysis of the steady-state distribution of thousands of native proteins within a sample. Multivariate data analysis is conducted using the MS proteomics packages MSnbase[29] and pRoloc[30], which provide a robust framework for processing, visualisation, and interrogation of spatial proteomics data as part of the open-source, open-development Bioconductor[31,32] suite of R software[33]. Additional modalities for phenotype discovery[34,35], transfer learning from heterogeneous data sources[36], assessment of cluster separation as a data resolution metric[37] and probabilistic classifiers such as Bayesian mixture modelling[38] have recently been integrated into the pRoloc pipeline.

HyperLOPIT provides a global snap-shot of an entire dynamic cellular system in an unbiased and untargeted manner. This methodology has previously been used for spatial mapping of the E14TG2a mouse embryonic stem cell line[21], the human U-2 OS osteosarcoma cell line[24,25], the yeast Saccharomyces cerevisiae[22,23] and apicomplexan parasite Toxoplasma gondii[27], with excellent resolution. Adapting spatial mapping methods such as hyperLOPIT and LOPIT-DC to track relocalisation events in response to perturbation or stimulation is now a major line of interest in the field and will greatly improve our understanding of dynamic biological processes and disease aetiology[39].

In this work we present a high resolution, global interrogation of the temporal and spatial proteome during an innate immune response in THP-1 cells, making use of the enhanced resolution of hyperLOPIT to capture protein relocalisation upon stimulation with LPS. The resulting data are showcased in a freely available interactive R Shiny application (http://proteome.shinyapps.io/thp-lopit/).

## Results

**Temporal proteome analyses reveal a cell-wide pro-inflammatory response in THP-1 cells during 0–24 h of LPS stimulation.** The experimental and computational workflows for this study are outlined in Fig. 1. Functional biological processes can be independently regulated both by changes in protein abundance and also by protein trafficking and translocation. Therefore, it was important to first assess the global proteomic landscape of the THP-1 monocytic leukaemic cell line within our time frame of interest. Total (unfractionated) cell lysates were collected at various time points following LPS stimulation (0, 2, 4, 6, 12, and 24 h) and processed for quantitative, MS-based proteomics analysis. 4292 proteins were reproducibly quantified in three biological replicate experiments, at all 6 time points (Supplementary Data 1, Supplementary Fig. 1a). A total of 311 proteins were found to be altered in abundance during the time-course of LPS stimulation, with statistical significance (adjusted $p$-value < 0.01 and $\log_2 FC > 0.6$) (Supplementary Data 1). The functional effects of these changes are outlined below.

**The early phase response to LPS (2–6 h) involves activation of classic pro-inflammatory mediators and an antioxidant response.** Interleukin 1-beta (IL1B) was the only protein found to be significantly upregulated in expression by 2 h-LPS and remained elevated in abundance throughout the 24 h time-course (Figs. 2a, e). Upregulated cellular levels of IL1B correlated with increased IL1B secretion into the cell supernatant, as measured by ELISA (Fig. 2d). By 4 h-LPS, three additional proteins were upregulated, all of which are characteristic of an early pro-inflammatory IFN response (IFIT2, IFIT3, CXCL10) (Fig. 2b). The increase in the intracellular levels of the chemokine CXCL10 coincided with its elevated secretion, as measured by ELISA (Fig. 2d).

Thirty-two proteins were significantly altered at the protein level by 6 h-LPS (Fig. 2c), 17 of which are known IFN-inducible,

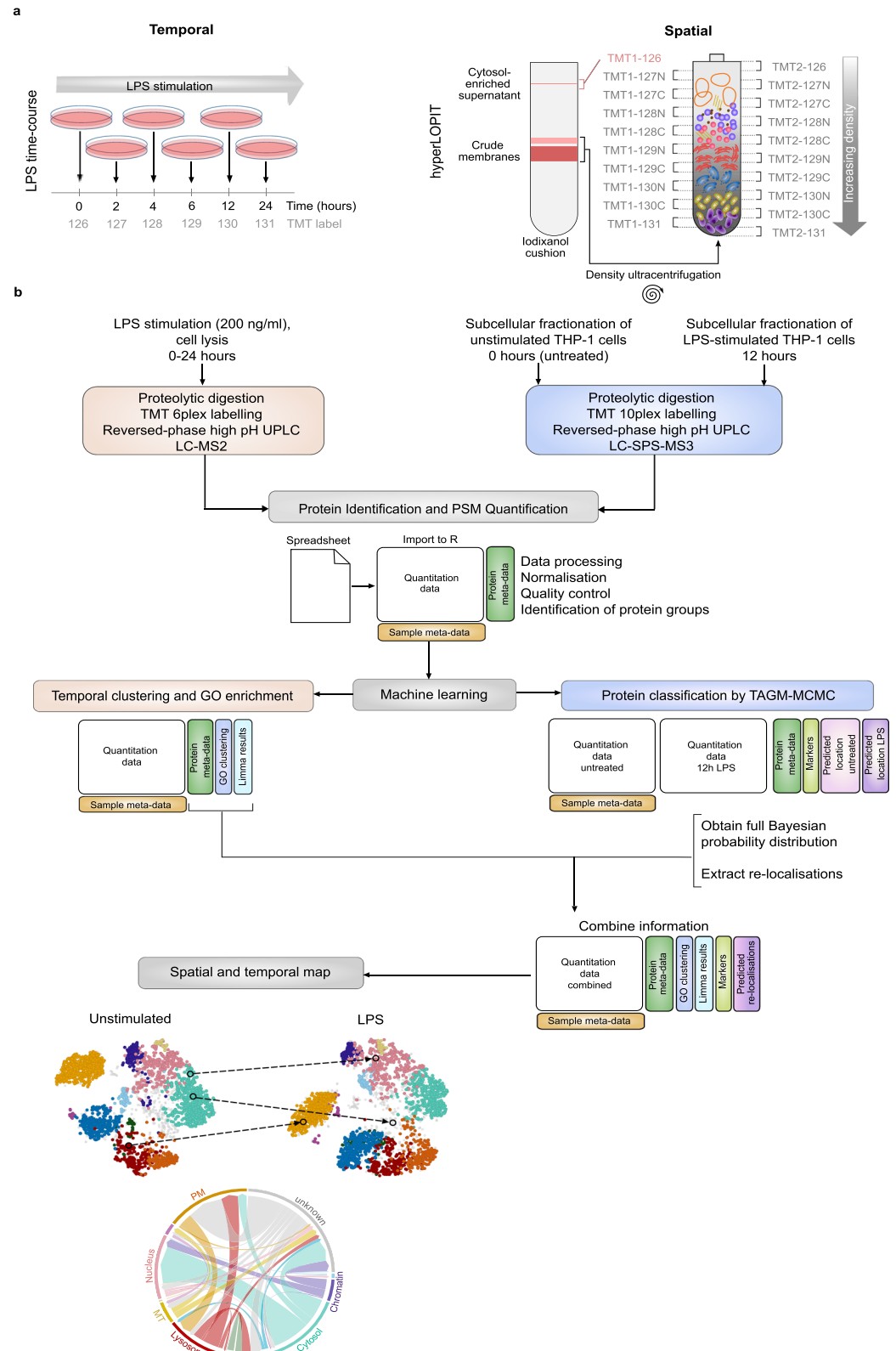

**Fig. 1 Overview of the spatiotemporal experimental pipeline.** **a** TMT labelling strategy for the temporal and the spatial MS proteomics analysis. Each experiment was performed in triplicate. **b** Experimental and computational workflows.

anti-microbial response proteins. Upregulation of PARP14 at 6 h-LPS supports the existing evidence that PARP14 is strongly induced by LPS stimulation and controls a subset of type 1 IFN-inducible proteins[40] including the heterodimers DTX3L–PARP9[41]

and IFI35–NMI[42], all of which were also found to be elevated in abundance during the time-course.

The generation of reactive oxygen species (ROS) can also drive a pro-inflammatory macrophage response upon LPS stimulation.

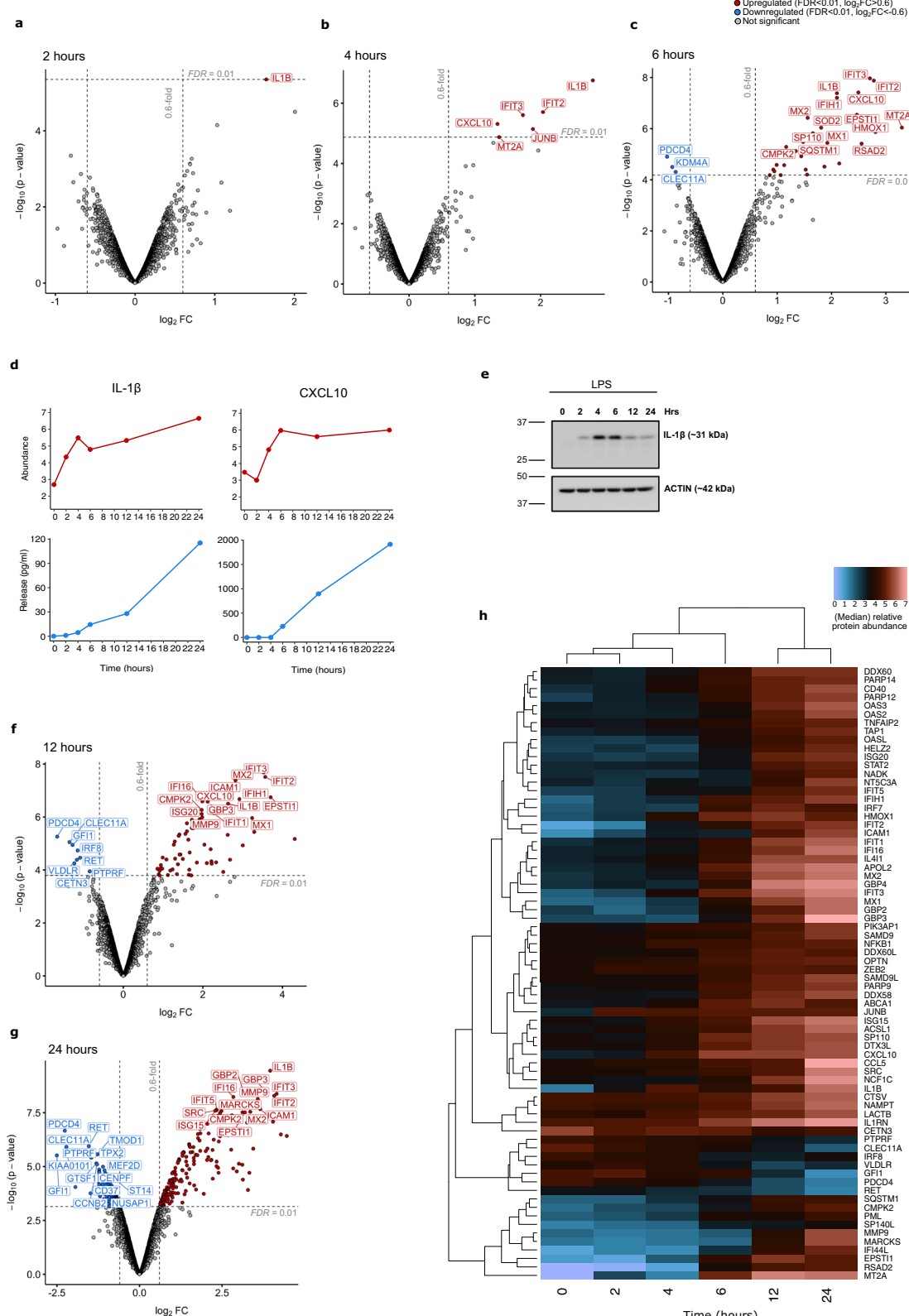

By 6 h-LPS, a proteome-wide response to increased intracellular ROS was apparent, through the elevation of the anti-oxidative superoxide dismutase 2 (SOD2), myeloid leukaemia cell differentiation protein (MCL1) and haem-oxygenase-1 (HMOX1). These findings agree with Widdrington et al.[43], who demonstrated that exposure of THP-1 monocytes to LPS resulted in a significant increase in ROS production by 6 h, triggering the induction of

antioxidant defense, mitophagy and mitochondrial biogenesis programmes.

There was concomitant downregulation of three proteins at the 6 h-LPS time-point, CLEC11A, a haematopoietic growth factor that acts as a secreted cytokine during monocyte/macrophage recruitment[44], KDM4A, oxygen sensing histone demethylase, and the tumour suppressor PDCD4.

**Fig. 2 Time-course of quantitative abundance changes in proteins during 24 h of LPS stimulation. a** Volcano plots showing the abundance of all 4292 time-course proteins at 2 h, **b** 4 h, **c** 6 h of LPS stimulation. **d** Median quantitative values for intracellular proteomics abundance (red) and extracellular, cytokine secretion levels as measured by ELISA (blue) for IL1B and CXCL10 expression ($n = 3$ biologically independent experiments). **e** Western blot ($n = 1$) of IL1B expression in THP-1 lysates during 24 h-LPS stimulation. Molecular weight markers in kilodaltons (kDa). The same blot was stripped and re-probed with anti-SQSTM1 for Fig. 6c. The same actin blot was therefore also used as a loading control. **f** Volcano plot demonstrating 72 proteins significantly changed in abundance by 12 h-LPS. **g** Volcano plot demonstrating 309 proteins significantly changed in abundance by 24 h-LPS. **h** Heatmap of the protein abundance for the 72 proteins significantly changed in abundance by 12 h-LPS during the time-course analysis. For all volcano plots proteins that are significantly upregulated (FDR < 0.01, $\log_2$FC > 0.6) and downregulated (FDR < 0.01, $\log_2$FC < 0.6) are highlighted in red and blue, respectively. Where relevant the top 15 most significantly changed proteins in each direction have been annotated with their corresponding gene name. The source data are provided as a Source Data file.

**Delayed THP-1 response to LPS reveals an expansion in anti-microbial immunity and the initiation of a regulatory anti-inflammatory process.** Twelve hours after the initial exposure to LPS, 72 proteins were identified as having been altered in abundance, with 64 upregulated and 8 downregulated (Fig. 2f, h). The immune response at 12 h-LPS continued to be dominated by an elevation in proteins known to be activated by LPS, including CCL5 (RANTES), RIG-I (DDX58), OAS2 and IFIT5. There was also evidence of the initiation of an anti-inflammatory process at 12 h-LPS with the increase in expression of NT5C3A, ABCA1, IFI44L, SAMD9, and MARCKS, all of which have been shown to inhibit the pro-inflammatory immune response. Interestingly, THP-1 cells appear to have adapted to the prolonged LPS stimulus at 12 h by modulating proteins that bind and transport LPS. A reduction in the expression of very-low-density lipoprotein receptor (VLDLR) and elevation of phospholipid-transporter ABCA1 was apparent by 12 h-LPS. VLDLR[45] and ABCA1[46] have been shown to control the import and export of LPS, respectively.

There was a significant reduction in the abundance of eight proteins by 12 h-LPS, including the zinc finger protein GFI1, known to regulate the LPS driven TLR4 response by antagonising NFkB activity in THP-1 cells[47]. The proto-oncogene tyrosine kinase receptor RET was also down-regulated by 12 h-LPS, which was previously observed to be reduced in expression during differentiation of THP-1 cells towards the macrophage lineage[48]. Furthermore, the elevation of a classical marker of M1 macrophage differentiation, epithelial–stromal interaction protein 1 (EPSTI1)[49] at 6 h-LPS, and the upregulation of intercellular adhesion molecule ICAM1 at 12 h-LPS, further supports the view that THP-1 cells are primed by 12 h-LPS to undergo a polarisation towards a pro-inflammatory phenotype.

**Cell cycle arrest, altered cellular morphology and upregulation of proteins associated with antigen presentation and T cell stimulation are apparent by 24 h of LPS exposure.** The THP-1 proteome demonstrates a major alteration by 24 h-LPS, with 178 upregulated and 131 downregulated proteins (Fig. 2g). Monocyte to macrophage maturation occurs upon LPS exposure and the substantial change to the proteome at 24 h clearly reflects this transition between these two phenotypes. We identified upregulation of CD14, the main co-receptor for LPS which mediates TLR4 signalling upon LPS binding, as well as several signalling and scaffold molecules involved in the TLR4 response, including SASH1, THEMIS2 and the scavenger receptor CD36. The IFN response was sustained at 24 h-LPS, with the additional upregulation of anti-microbial molecules such as OAS1, EIF2AK2, TRIM14 and CMTR1. Maturation towards a pro-inflammatory macrophage phenotype was also evident by the upregulation of proteins associated with antigen presentation and peptide loading at the endoplasmic reticulum (ER) (HLA-A HLA-B, B2M, TAP1, TAP2) and T cell co-stimulation (ICAM1, ADA, CD40 and SPN). The monocytic adhesion molecule ITGAX (CD11C) is known to be down-regulated during the monocyte-to-macrophage

transition. Our proteomics data demonstrate a loss in ITGAX expression at 24 h-LPS, suggesting that the process of macrophage maturation is underway at this time point. Macrophage polarisation by 24 h is further supported by the upregulation of PLEKHO2, which has previously been linked to macrophage survival, differentiation and maturation in mice[50].

A substantial number of downregulated proteins at 24 h-LPS are associated with cell division, DNA replication, centrosomal, microtubule and kinetochore regulation (Supplementary Data 2). A reduction in cell cycle proteins corresponds to the previously reported cell cycle arrest upon LPS stimulation of THP-1 cells[51] and a reduced expression of the classic marker of cellular proliferation, MKI67, was observed.

A number of molecules known to be involved in cytoskeletal processes including the regulation of actin dynamics, extracellular matrix (ECM) and cellular migration were altered in abundance (Supplementary Data 2). Downregulation of lymphocyte-specific protein (LSP1) was detected, which is a critical regulator of actomyosin contractility and migration in primary macrophages[52]. The proteoglycan serglycin (SRGN) was also reduced and has been previously shown to decrease during the monocyte-to-macrophage maturation of THP-1 cells[53].

By 24 h, the response to LPS appears to be cellular-wide with alterations in intracellular vesicular and endosomal trafficking, autophagy-lysosomal machinery, and metabolic regulators (Supplementary Data 2). LACC1 (FAMIN), a central regulator of the metabolic function of macrophages, plays a role in inflammasome activation, ROS production and bactericidal activity[54]. Components of the inflammasome such as CASP1, CASP4 and Pyrin are upregulated, which coincides with the continued secretion of IL1B. Together, our results demonstrate a global response to LPS stimulation in the THP-1 proteome over the course of 24 h, which is distributed throughout the subcellular landscape and coincides with the phenotypic switch from monocyte to macrophage.

**An overview of global proteomic changes using unsupervised Bayesian clustering in combination with annotation enrichment.** The proteomic time-course series was modelled using a Bayesian mixture model to capture clusters of co-regulated proteins and temporal patterns of protein expression in response to LPS. Replicate experiments were combined using multiple dataset integration (MDI)[55] where the number of clusters was automatically inferred. Using this unsupervised approach, 1057 proteins in the time-course series were assigned to 49 distinct clusters (Supplementary Data 3) and Gene Ontology (GO) enrichment analysis was used to identify overrepresented annotation terms associated with each cluster (Supplementary Data 4). Of the clusters, 17 were enriched for significantly overrepresented biological process (GOBP) related terms (adjusted $p$-value < 0.05), providing a functional insight into the biological activities affected by LPS within 24 h of stimulation.

Six representatives upregulated (Fig. 3a) and six downregulated (Fig. 3b) clusters are shown with corresponding GO annotation enrichments (Fig. 3c–e). Temporal Clusters One ($n = 46$), Two ($n = 103$) and Three ($n = 87$) all demonstrated an upregulated trend throughout the time-course (Fig. 3a), occurring at an early (4–6 h), mid (12 h) and late (24 h) stage, respectively. GO annotation enrichment confirmed that Cluster One was significantly enriched for the term "response to external biotic stimulus", Cluster Two was enriched for "innate immune response" and Cluster Three for "endocytosis" (Fig. 3a). Clusters Four ($n = 14$), Five ($n = 29$) and Six ($n = 58$) were also upregulated through the time-course and were enriched for "interspecies interaction between organisms", "tRNA transport" and "intracellular protein transport", respectively (Fig. 3c), further supporting a role for intracellular trafficking and translocation in response to LPS by 24 h.

Several clusters featured down-regulated trends throughout the LPS time-course (Fig. 3b). Clusters Nine ($n = 44$), Ten ($n = 10$) and Eleven ($n = 34$) were functionally associated with the GOBP terms "chromosome organisation", "sister chromatid segregation" and "cellular respiration", respectively, reinforcing the observation that DNA replication and cell proliferation were reduced during the progression of the time-course. The terms "RNA splicing" and "cellular response to stress" were overrepresented in Cluster 13 ($n = 37$) and Cluster 14 ($n = 105$), while "oxidation–reduction process" was the top-scoring GOBP term for Cluster 17 ($n = 76$) (Fig. 3c).

**The hyperLOPIT spatial proteomics platform allows subcellular interrogation of the THP-1 proteome at 0 h (untreated) vs. 12 h LPS exposure.** The time-course of LPS stimulation provides a holistic overview of protein expression in the THP-1 proteome during the initial phases of a pro-inflammatory response. By understanding protein relocalisation events, we gain a more complete picture of the cellular response to LPS. The 0 and 12 h LPS time-points were selected for deeper, spatial proteomics analysis of dynamic trans/relocalisation events occurring between subcellular regions (where 0 h represents untreated cells). These time points allow us to compare LPS effects in cells that had undergone substantial proteomic changes whilst retaining a non-adherent phenotype.

Three replicate experiments for untreated (0 h-LPS) and also for stimulated (12 h-LPS) THP-1 cells were performed according to the established hyperLOPIT methodology. The resulting spatial proteome maps consisted of 3882 and 4067 proteins reliably identified across all three replicates in the 0 h-LPS and 12 h-LPS conditions, respectively (Supplementary Data 5, Supplementary Fig. 1b, c). A subset of 3288 proteins was common across both conditions (Supplementary Data 6, Supplementary Fig. 1d), of which 78.5% were also identified in the unfractionated time-course experiment, providing both temporal and spatial data for 2581 proteins (Supplementary Fig. 1e). Single time-point spatial information is also available for proteins that demonstrated dramatic changes in expression, such as PDCD4, the most significantly down-regulated protein in the time-course, which was only found in the unstimulated hyperLOPIT condition. In contrast, IFIT1 and EPSTI1, two of the most significantly upregulated proteins, were only identified in the LPS-stimulated dataset. Furthermore, four pro-inflammatory cytokines shown to be secreted after LPS stimulation (IL8, IL1B, CXCL10, TNF) were identified with low abundance in the 12 h-LPS stimulated hyperLOPIT dataset and were absent in the 0 h-LPS data (Supplementary Fig. 1f).

**TAGM classification integrates subcellular localisation and the dynamic response to LPS stimulation.** A T-Augmented Gaussian Mixture Model with Bayesian computation performed

using Markov Chain Monte Carlo (TAGM-MCMC)[38], was used to confidently assign proteins to distinct subcellular regions and to capture and quantify the uncertainty in the allocation of proteins to these compartments. Organelle classification within the 3288 hyperLOPIT subsets was performed using 783 organelle marker proteins representing 11 subcellular niches (Fig. 4a, Supplementary Data 6 and 7). A total of 1717 proteins in the unstimulated dataset and 1713 proteins in the LPS dataset were classified to distinct subcellular regions with high confidence (Fig. 4b, Supplementary Data 6). The remaining proteins were classified as "unknown" or unannotated locations, due to either moonlighting proteins existing in mixed locations, trafficking proteins or uncertainty in discrete localisation. There was a high degree of correlation between unstimulated and stimulated datasets, with 61% of the identified proteome sharing the same organelle localisation in both conditions (75% including proteins labelled as unknown) (Supplementary Fig. 1g).

The TAGM classification results were confirmed by manual curation using the UniProtKB database. For example, 91% of proteins classified to the cytosol contained predicted UniProt annotations as "cytosol" or "cytoplasm". Of the total mitochondrial proteins, 95% had documented mitochondrial annotations, reaffirming the accuracy of the TAGM localisations. The lysosomal compartment contained 188 assignments common to both conditions, of which 104 were lysosomal resident proteins and 25 were endosomal. Interestingly, SNAP29, CD93, RAB21 and RAB23 were only associated with lysosomes in the LPS stimulated condition and are known to play a role in autophagosome formation, a process which is known to be underway in macrophages within 24 h of LPS stimulation[56].

LPS exposure appears to induce a dynamic alteration in the plasma membrane (PM). A total of 135 proteins were assigned to the PM in both conditions, of which 95% had UniProtKB PM annotations. CD14, the classic co-receptor for LPS, was localised to the PM in unstimulated conditions, however was no longer associated with the PM with high probability following stimulation. Although the LOPIT analysis does not indicate that this protein is undergoing relocalisation at 12 h-LPS (see Supplementary Fig. 1h), the data does suggest that the receptor may be decoupled from the plasma membrane at this time-point. This is not due to a reduction in abundance, as the protein is unaltered at 12 h-LPS and upregulated at 24 h-LPS, in the time series. CD14 is essential for TLR4-TRIF-mediated IFNs response through endocytosis of the signalling complex[57], which could account for the loss of CD14 at the PM and the induction of the IFN response seen in the time-course data.

**LPS stimulation results in protein relocalisation events and reveals a highly dynamic proteome.** Based on the combination of joint posterior and outlier probabilities, protein trans/relocalisation events that occur following LPS stimulation were classified as four distinct scenarios: (i) Type 1: organelle-to-organelle, (ii) Type 2: unknown-to-organelle, (iii) Type 3: organelle-to-unknown, and (iv) Type 4: unclassified proteins that exhibit large changes in their posterior probability distribution between conditions. For additional stringency and to capture large movements in probability space, all potential relocalisation events were ranked according to their proteins natural L2 distance between their TAGM joint posterior probabilities. A large L2 distance between probabilities is indicative of proteins that exhibit large movements in probability space. High confidence movements were identified for 253 proteins out of the 3288 proteins analysed, of which 112, 62, 49 and 30 were assigned to Type 1–4 translocation events, respectively (Supplementary Data 8). A literature search revealed that of these 253 proteins, 92 (36%) had been

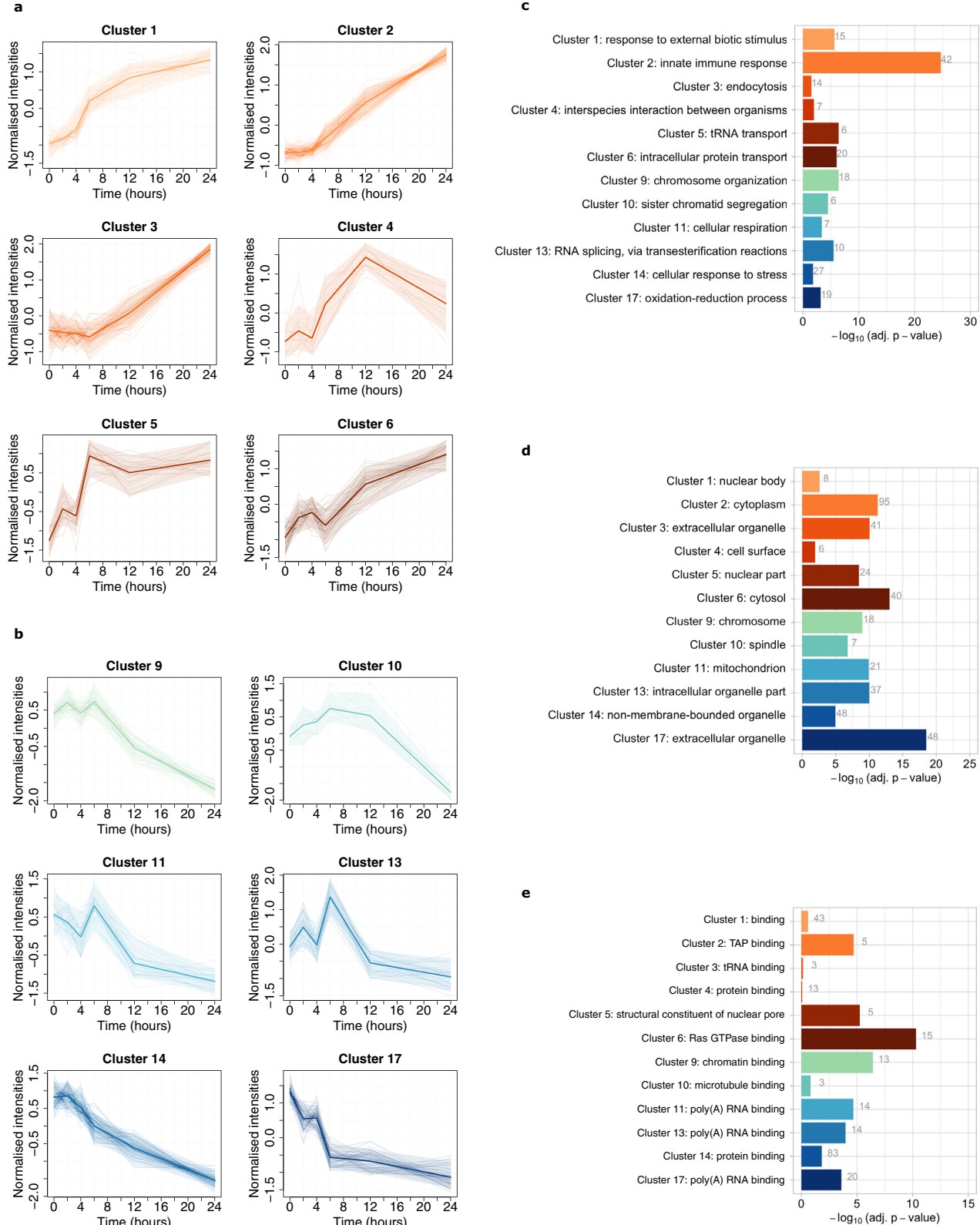

**Fig. 3 Bayesian temporal clustering of the LPS time series demonstrates clusters of co-regulated proteins. a** Shown are multiple dataset integration (MDI) profiles for each protein found in 6 representative upregulated and **b** 6 downregulated clusters from the Bayesian temporal clustering analysis. The single bold line profile on each plot shows the mean normalised MDI profile and the 0.05 and 0.95 quantiles of each cluster are highlighted by shaded bands. **c** Gene Ontology Biological Process (GOBP) **d** Cellular Component (GOCC) and **e** Molecular Function (GOMF) annotation term enrichment for the clusters are shown. X-axis: –log$_{10}$(adj.p-value). The numbers within the bar charts refer to the number of proteins associated with that term in each cluster. Source data are provided as a Source Data file.

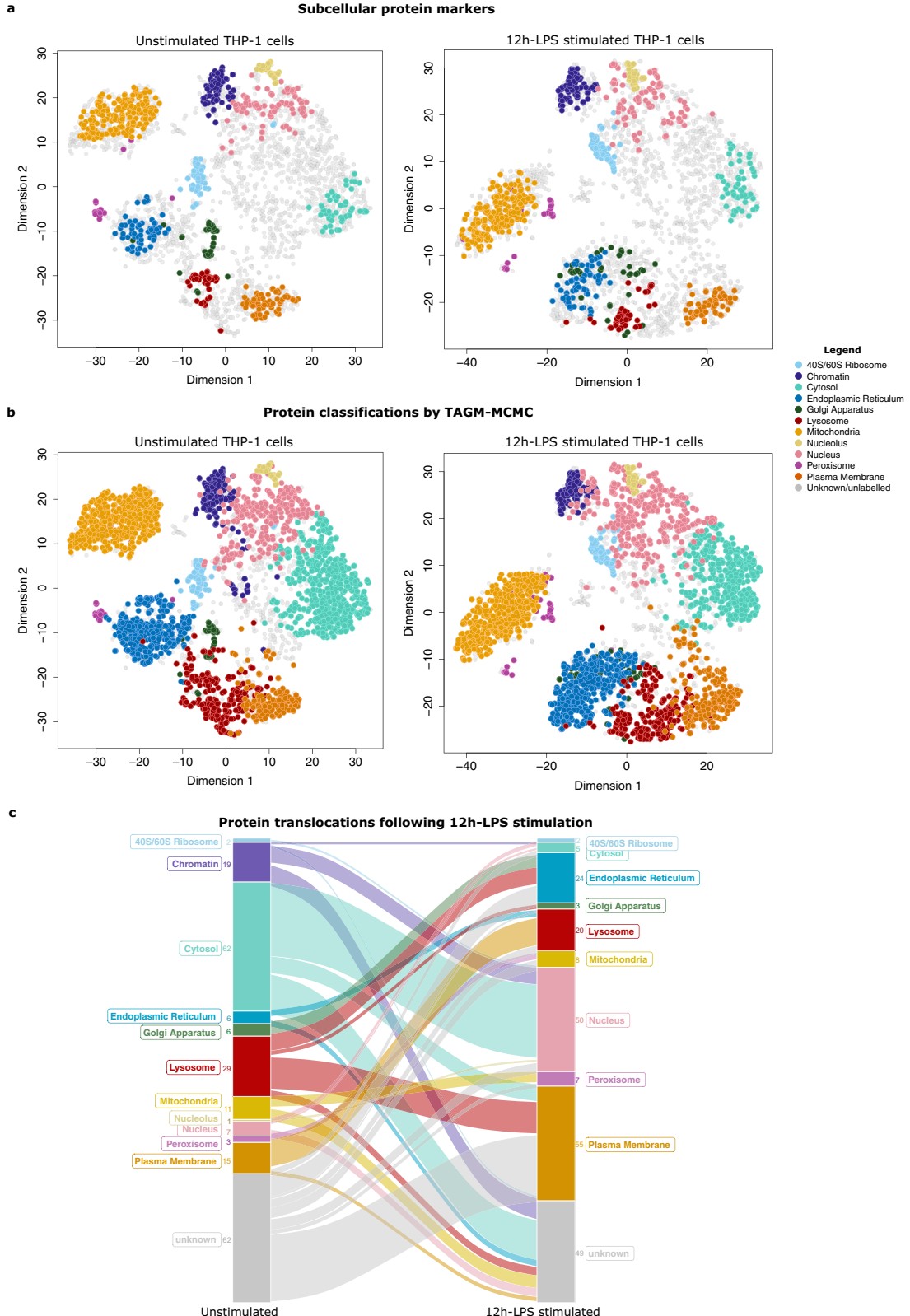

**Fig. 4 Assignment of proteins to subcellular organelles using TAGM-MCMC semi-supervised classifier for spatial proteomics data. a** Shown are the t-SNE projections of the unstimulated (0 h-LPS) and stimulated (12 h-LPS) hyperLOPIT datasets. 3288 proteins are shown in each plot and coloured by organelle protein marker classes. Proteins of unknown/unlabelled locations are shown in grey. **b** Shown is the same t-SNE projections following TAGM-MCMC classification. **c** An alluvial plot representing the 253 proteins that are found to move between organelles at 12 h-LPS. The number of proteins that are found to translocate to and from each subcellular compartment is denoted next to the labelled strata. The colours of organelles are matched to the hyperLOPIT organelle colour key.

previously identified as being directly involved in innate immunity and 51 (20%) were linked specifically with the TLR4/LPS response (Supplementary Data 9). The remaining 160 (64%) proteins that underwent translocation in response to 12 h-LPS have not been previously reported to be influenced at the protein level by TLR4/LPS stimulation. The majority of these proteins are involved in cytoskeletal remodelling, the endolysosomal system and nucleo-cytoplasmic shuttling, processes known to be influenced by LPS stimulation. GO annotation enrichment analysis confirmed that protein localisation (GO:0008104~protein localisation) and transport (GO:0015031~protein transport) are highly significant functional processes for this group of 253 relocalising proteins (see Supplementary Data 11).

**The effect of LPS stimulation on protein localisation is variable across subcellular compartments.** By plotting the 253 LPS-induced relocalisation events onto the hyperLOPIT t-SNE projections, it became clear that these proteins were distributed throughout all subcellular compartments (Supplementary Fig. 2a). The dynamic proteome with the directionality of movement can be represented on an alluvial plot (Fig. 4c).

Many translocation events occurred in the cytosol for example, with 67 proteins found to be entering (5/67) or leaving (62/67) this compartment (Fig. 4c, Supplementary Fig. 2b). Overall, 11% of all classified cytosolic proteins were found to change location after LPS stimulation. Proteins were found to move from the cytosol to either the nucleus (35 proteins) or the PM (8 proteins), with a further 19 moving to an unclassified location. Only five proteins were found to become cytosolic after 12 h of LPS stimulation.

Substantial changes in the proteome composition were also identified in the nucleus and lysosome (Supplementary Fig. 2c, d) with 57 and 49 translocators, respectively. Our analysis also revealed a dynamic regulation of the endomembrane system following LPS stimulation, with 41 proteins in the process of being shuttled between the ER, Golgi apparatus, lysosome and unknown compartments. Conversely, the mitochondria proved to be a relatively stable organelle at 12 h-LPS, with only 23 proteins undergoing mitochondrial import or export, corresponding to 4.5% of all mitochondrial assignments. The hyperLOPIT analysis, therefore, provided evidence of a highly dynamic proteome, which affects each subcellular compartment differently. A selection of these translocation events is highlighted below.

**Nucleo-cytoplasmic shuttling plays a role in LPS-induced signalling.** Thirty-five proteins were found to translocate from the cytosol to the nucleus, whereas only two proteins moved from the nucleus (TRAF2) or the chromatin (DENND4B) to the cytosol. The cytosol-to-nucleus group contained a nuclear importin (KPNA2), nuclear exportin (XPO5) and nucleolin (NCL), which can play a role in nucleocytoplasmic transport of newly synthesised pre-RNAs, as well as 11 subunits of the eukaryotic translation initiation factor 3 (EIF3) complex and 4 members of the EIF4 complex (Supplementary Fig. 3a). These complexes mediate the recruitment of ribosomes to mRNA.

The hyperLOPIT data was sufficiently well-resolved to identify 9 sub-organellar relocalisation events within the nucleus (Supplementary Fig. 3b) (1 nucleolus-to-nucleus and 8 chromatin-to-nucleus). We observed the relocalisation of CTDSPL2 from the chromatin to the nuclear cluster following LPS stimulation. CTDSPL2 can be released from transcriptionally silenced chromosomal regions during erythroid differentiation[58] and it has been associated with a number of processes identified in this study, including cellular adhesion, metabolism and protection from oxidative stress-induced apoptosis.

**LPS stimulation results in protein reorganisation at the PM.** The PM is known to undergo a massive structural reorganisation during monocyte-to-macrophage polarisation and differentiation. This region was the most dynamically regulated organelle in the hyperLOPIT study, with 70 proteins found to translocate to (55/70) or from (15/70) the PM in response to LPS, accounting for 27% of all the identified PM proteins (Supplementary Fig. 3c). There was a transfer of proteins between the PM and lysosomal compartment, with 13 proteins moving from the PM to the lysosome, and 15 travelling in the opposite direction. Among the lysosome-to-PM group of proteins were several known to be involved in the LPS/TLR4 response, namely CCR1, SLC38A2, PSEN2, DAB2 and MSN. This group also included ABHD17A, a depalmitoylase involved in the dynamic regulation of protein localisation and signalling[59] and RNF167, a trafficking, endocytic E3 ubiquitin-ligase which regulates lysosomal exocytosis and PM resealing[60].

Of the eight proteins which translocated from the cytosol to the PM at 12 h-LPS, five have been reported to be involved in LPS/TLR4 regulation, including CLIC4 which associates with pro-inflammatory cytokine secretion following LPS stimulation[61] and translocates from the cytosol to the PM in murine macrophages exposed to LPS[62], and HGS, which plays an important role in the initial steps of TLR4 sorting to the endosomal pathway following engagement of LPS[63]. Other proteins in this group, such as CDC42, are discussed below.

**Induction of autophagosome formation and activation of lysosomal degradation pathways are apparent at 12 h LPS.** Autophagy machinery can be recruited to the site of the phagophore following stimulation, where it initiates the formation of the autophagosome, which subsequently fuses with the lysosomes to form autophagolysosomes. Autophagy receptors sequestosome (SQSTM1/P62) and CALCOCO2 (NDP52), as well as TOM1, an endosomal protein and binding partner of CALCOCO2, undergo LPS-induced translocation to the PM. We observed that TOM1, SQSTM1 and CALCOCO2 co-localise in both unstimulated and stimulated datasets suggesting they may be moving together as a complex (Supplementary Fig. 3d). TOM1 has been shown to interact directly with another endosomal trafficking molecule, TOLLIP[64], which moves from the cytosol to the PM in hyperLOPIT.

The scavenger receptor class B member 1 (SCARB1) and LRCH4 both relocate from the PM to the lysosome. SCARB1 was shown to protect against sepsis by promoting LPS clearance in hepatic cells[65] and to suppress TLR4-LPS signalling in mouse macrophages[66]. LRCH4 has recently been identified as a TLR4 accessory protein with a role in the regulation of the early phase of the innate immune response to LPS, by promoting the docking and delivery of LPS to the lipid raft membranes in the vicinity of CD14[67].

**Evidence of relocalisation of Rho-GTPases after 12 h LPS stimulation.** Cytoskeletal alterations are apparent at the subcellular level by 12 h-LPS, with the movement of a number of key intracellular cytoskeletal regulators such as the Rho-GTPase family members CDC42, RAC1, RHOA, and RHOC (Fig. 5a, b). Confocal microscopy demonstrated that CDC42 undergoes a redistribution from a diffuse cytoplasmic location to punctate spots in the cell periphery (Fig. 5d) despite being unchanged in protein abundance (Fig. 5c). Previous studies have shown that CDC42[68], RHOA[69] and RAC1[70] undergo LPS-induced GDP-GTP exchange and activation. Additional Rho-GTPase-accessory proteins were also observed to translocate to the PM in response to LPS stimulation (SRGAP2, ARHGAP18, APBB1IP) (Fig. 5a).

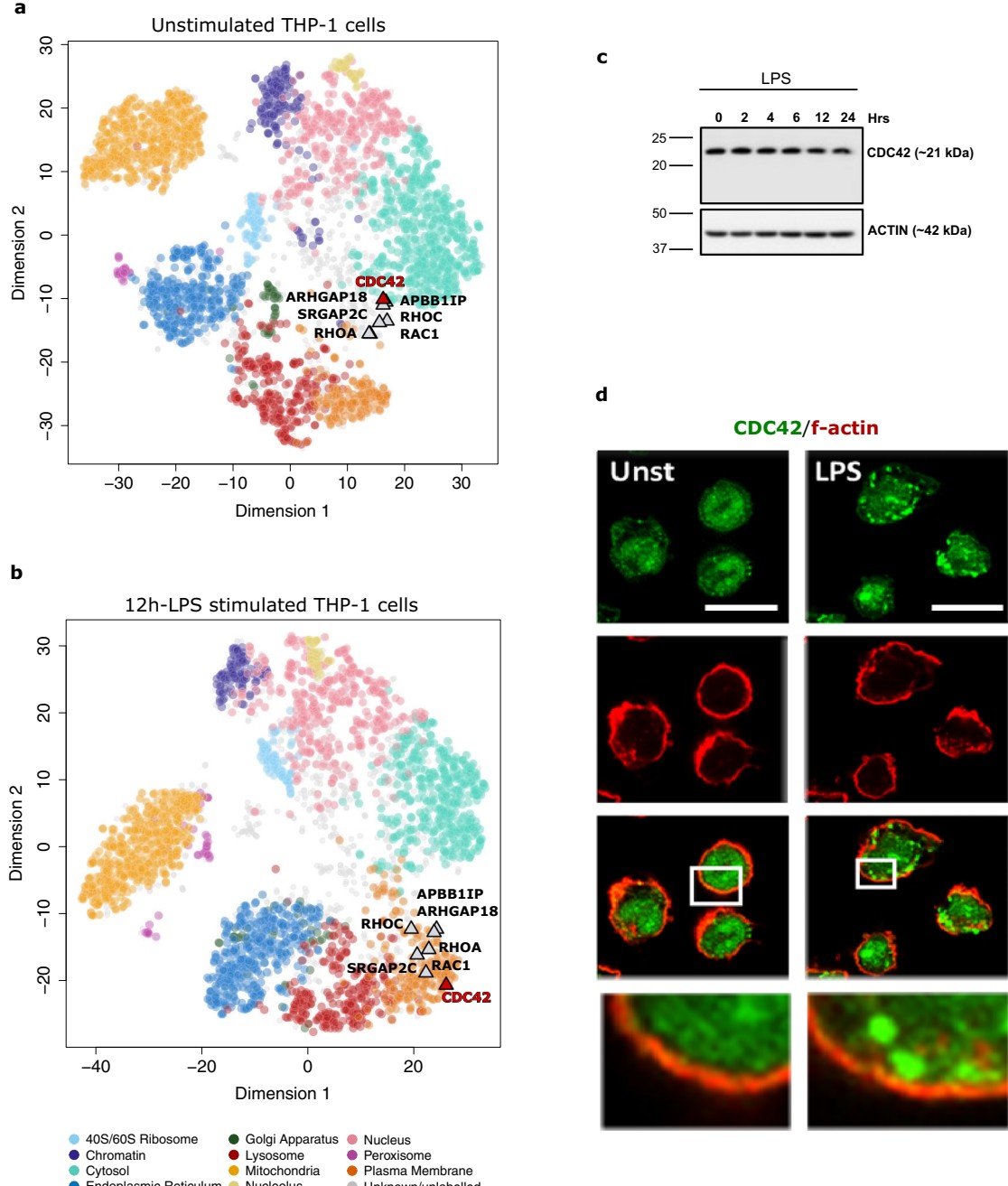

**Fig. 5 Relocalisation of RHO-GTPase trafficking molecules to the PM.** Relocalisation of RHO-GTPase family members and accessory proteins were found in response to LPS, including CDC42, RAC1, RHOA, RHOC, SRGAP2, ARHGAP18, APBB1IP. These proteins are shown in **a** unstimulated and **b** LPS-stimulated conditions. **c** Western blot of CDC42 in the 24 h-LPS time-course. Anti-actin western blot is shown as control. Molecular weight markers are indicated in kilodaltons (kDa). **d** Confocal image (×63) of THP-1 cells ± LPS for 12 h and labelled with anti-CDC42 (Green) and phalloidin (Red). Scale bar = 10 μm. All western blots and images were conducted at n = 1. Source data are provided as a Source Data file.

SRGAP2 is recruited during leucocyte attachment to direct cell polarisation and adhesion[71]. APBB1IP (RIAM) is a signal transducer involved in actin cytoskeletal remodelling during complement-dependent phagocytosis in THP-1 cells, a process significantly enhanced after LPS stimulation[72].

**Translocation of vesicular trafficking molecules regulate the intracellular response to LPS.** The hyperLOPIT analysis identified the translocation of many vesicular trafficking molecules (Supplementary Fig. 4a), including the synaptic vesicle glycoprotein 2A (SV2A), the trafficking protein TRAPPC3, the

clathrin-associated AP-1 complex subunit gamma (AP1G1), the WASH complex subunit strumpellin and the endosomal charged multivesicular body protein (CHMP2B). CHMP2B is a member of the ESCRT-III complex (endosomal sorting complex required for transport) and moved from an unknown location to the lysosome. Several other members of the ESCRT-III complex co-localised with CHMP2B in the lysosome of the LPS-stimulated condition (Supplementary Fig. 4c), although they did not meet the threshold to qualify as movers. Furthermore, IST1, which moved from the cytosol to an unannotated location, is also involved in the ESCRT III complex and in endosomal fission at

sites of ER-endosomal contact, but did not localise with the CHMP2B complex. SYTL4 and SCAMP2 underwent relocalisation from the PM to the lysosome, both of which are involved in recycling secretory vesicles and granular exocytosis.

The RAB family of small GTPases act as master regulators of vesicular trafficking and modulators of the immune response. The hyperLOPIT analysis revealed relocalisation of 12 RABs (Supplementary Fig. 4b), eight of which translocated to the ER, possibly reflecting a role in antigen presentation or trafficking of newly synthesised molecules. RAB23, which has previously been shown to facilitate phagolysosomal fusion[73], moved from the PM to the lysosome, whereas RAB6B moved from the lysosome to the PM. Several of these RAB traffickers are known to be associated with the LPS/TLR4 immune response (Supplementary Data 9) and others are involved in autophagosome formation, including RAB7 and RAB32.

**Several proteins undergo both spatial and temporal modulation in response to LPS.** Of the hundreds of proteins which were altered during either the 24 h-LPS time-course or the hyperLOPIT analysis, only 15 proteins were found to change both their location (at 12 h) and their total protein abundance level in response to LPS over the 24 h time period (Fig. 6a, b, e). Of these, only the autophagy receptor SQSTM1 and CLEC11A were changed in abundance at the 12 h-LPS time-point. SQSTM1 became more abundant (Fig. 6c) and also trafficked to the PM. This multifunctional autophagy receptor is known to be upregulated by LPS[74]. Meanwhile, CLEC11A became less abundant (Fig. 6d) and relocated from the lysosome to the ER. We postulate that the reduced expression of CLEC11A, together with it being trafficked through the endomembrane pathway, may indicate that this cytokine molecule is actively secreted from THP-1 cells during LPS stimulation. This data indicates that distinct relocalisation events and abundance changes are both involved in the global, cellular response to LPS.

**HyperLOPIT provides a scaffold to visualise and interrogate proteins of interest.** Having established that hyperLOPIT provides a means to robustly classify unannotated proteins to distinct organelles and to investigate protein relocalisation events, we next interrogated the THP-1 hyperLOPIT dataset for visualisation of (i) Bayesian temporal clusters, (ii) protein complexes, (iii) protein–protein interaction partners, and (iv) specific subproteome populations identified in existing publicly available datasets.

Three clusters from the Bayesian temporal clustering analysis of the LPS time-course experiment were overlaid onto the hyperLOPIT spatial maps (Fig. 7a) revealing that temporal clusters can be spatially distributed and highly enriched within certain organelles. Clusters 9, 11 and 17 were enriched in the hyperLOPIT chromatin, mitochondria and cytosol, respectively.

Protein complexes are well preserved during the hyperLOPIT procedure (Fig. 7b), including the exocyst complex, actin-related protein 2/3 complex, coatomer proteins and the DNA replication licensing MCM complex. Furthermore, entire complexes were found to relocalise in response to LPS stimulation, as seen by the EIF3 complex which translocated from the cytosol into the nucleus (Fig. 7b, Supplementary Fig. 3a).

Protein–protein interaction partners are also preserved during the hyperLOPIT protocol (Fig. 7c). For example, we see co-localisation of (i) CDC42 and TRIP10 (CDC42-interacting protein 4)[75], (ii) the endocytic molecules HGS and TSG101[76] and (iii) the mitochondrial NLRX1 and FASTKD5[77]. Proteins known to co-localise upon LPS stimulation were also shown together in hyperLOPIT space, including IKBKG (Nemo) with

IKBKB[78]. The LPS-responsive interactors PARP9 and DTX3L[41] were both upregulated in abundance by 12 h-LPS and co-localised in hyperLOPIT space. Likewise, the interferon-inducible binding partners IFI35 and NMI[42] were co-localised and upregulated following LPS stimulation at 24h-LPS.

Finally, we used the hyperLOPIT dataset as a scaffold to examine additional spatial protein information identified in complementary proteomics studies. A secretome analysis of LPS-activated murine macrophages[79] was found to predominantly localise to the endomembrane secretory pathway in hyperLOPIT (Supplementary Fig. 5a). Similarly, a cell surface proteome study of THP-1 cells during differentiation into macrophage-like cells[11] maps closely to the PM cluster of hyperLOPIT (Supplementary Fig. 5b). A macrophage-derived RNA-binding protein (RBP) interactome[80] was enriched in hyperLOPIT nuclear, nucleolar, ribosomal, and cytosol locations (Supplementary Fig. 5c). A recent study which isolated the lysosomal and mitochondrial fractions from murine macrophages was found to overlap with the hyperLOPIT mitochondrial and lysosomal locations[81] (Supplementary Fig. 5d). Together, these results demonstrate that the hyperLOPIT approach can be used as a platform to overlay multiple layers of spatial proteomic information in order to gain a complete global perspective on highly complex cellular processes.

**Discussion**
The integrated proteomic analysis presented here has demonstrated that the pro-inflammatory innate immune response to LPS in the THP-1 monocytic cell line is regulated by both protein expression changes and subcellular relocalisation events. Combining temporal abundance profiling, hyperLOPIT spatial proteomics, and a state-of-the-art Bayesian analysis framework, we infer dynamic protein events within and between organelles during the LPS immune response. This study provides robust, quantitative, temporal data for 4292 proteins across a 24 h time-course of LPS stimulation, as well as the subcellular localisation of 3288 proteins in unstimulated and 12 h-LPS, stimulated conditions. This analysis has revealed the translocation (253 proteins) and altered abundance (311 proteins) of hundreds of proteins in response to LPS distributed across different subcellular compartments, with only 15 proteins undergoing changes in both abundance and location. Many of the proteins identified in this study are known to be involved in immune-related functional processes including a pro-inflammatory IFN response, PM reorganisation, autophagosomal induction, endolysosomal trafficking, cytoskeletal remodelling, and nucleocytoplasmic shuttling. Furthermore, several proteins presented here have never been associated with the proinflammatory response and may represent novel targets for therapeutic intervention. This spatial proteomics study represents the most extensive insight into the LPS-driven innate immune response reported to date, allowing the simultaneous localisation of thousands of proteins undergoing perturbation in a complex and dynamic system.

A cross-comparison of publicly available datasets shows that many LPS-regulated proteins in different cell types and conditions exhibit the same trend in our time-course quantitative abundance dataset (Supplementary Data 10) as well as in the hyperLOPIT list of translocating proteins (Supplementary Data 9), thereby validating many of our findings presented here. A number of previous studies have utilised various proteomics approaches to explore the LPS-induced innate immune response in macrophages[9,10,82,83], as well as to characterise individual subcellular organelles following LPS activation[11,12]. This study presents a global analysis to simultaneously track thousands of native proteins in response to LPS with a spatial context. Of note,

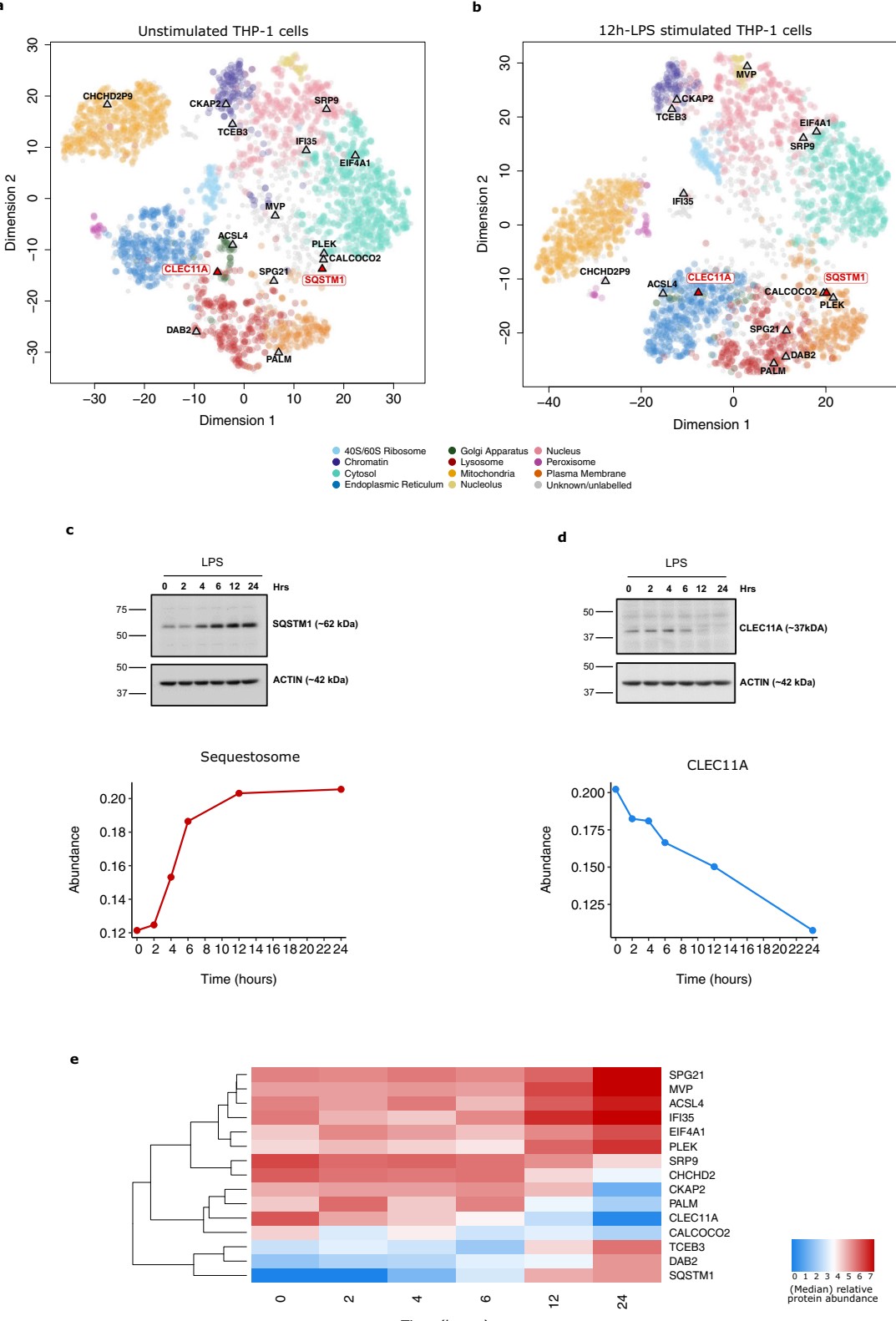

the interpretation of the exact mechanisms that underlie the observed relocalisation of proteins in our study is outside the scope of this study. It is not possible to distinguish trafficking of existing proteins from one location to another, from proteins being differentially degraded at one location and newly synthesised proteins locating to an alternate location. The time frames of our LPS treatment would enable both of the scenarios to occur.

We believe our data shows that observed 'movers' are functionally in different locations upon LPS treatment compared with untreated cells.

The field of spatial proteomics is rapidly developing and many powerful MS-based technologies now exist to provide insights into the investigation of organelle remodelling, controlled movement between subcellular regions and protein mis-

**Fig. 6 Proteins can exhibit both spatial and temporal regulation by LPS.** 15 proteins were altered in abundance during the time-course of LPS stimulation and also translocated to different subcellular regions. The distribution of these 15 proteins in hyperLOPIT space is shown for **a** the unstimulated and **b** the LPS-stimulated conditions. **c** Western blot and the median TMT reporter ion quantitation (red) for SQSTM1 in the 24 h-LPS time-course. Anti-actin western blot is shown as control. Molecular weight markers are indicated in kilodaltons (kDa). The immunoblot in Fig. 2e was stripped and re-probed for anti-SQSTM1. The same actin blot was therefore also used as a loading control. **d** Western blot and proteomic quantitation (blue) for CLEC11A in the 24 h-LPS time-course. Anti-actin western blot is shown as control. Molecular weight markers are indicated in kilodaltons (kDa). **e** A heatmap of the relative quantitative abundance values for these 15 proteins during the time-course experiment. All western blots were conducted at $n = 1$ and temporal proteomics at $n = 3$ biologically independent experiments. Source data are provided as a Source Data file.

localisation in multiple model systems (reviewed in refs. [13,14]). The hyperLOPIT platform and its derivative LOPIT-DC have now been applied to multiple biological contexts and questions[21,22,24,25,27,28]. We present here the first iteration of dynamic-hyperLOPIT for characterisation of protein relocalisation events during cellular perturbation using the THP-1 monocytic cell line which is notoriously difficult to use for microscopy imaging. We have shown that the LPS/TLR4 innate immune response in the THP-1 cell line is regulated both spatially and temporally. By releasing the hyperLOPIT dataset as an interactive resource for the scientific community, the data can be mined for individual proteins, biological complexes, pathways of interest, or interrogated with other publicly available datasets. Obtaining a greater understanding of the immune response to LPS will provide the opportunity to develop strategies in immune manipulation, which will be of benefit in the development of vaccines, anticancer therapies and disease treatments.

## Methods

**Cell culture and LPS stimulation.** THP-1 cells (ATCC® TIB-202™) were cultured in RPMI 1640 medium with GlutaMAX™ (Gibco), supplemented with 10% foetal bovine serum (Sigma), 100 U/mL of Penicillin and 100 µg/mL Streptomycin (Gibco), 20 mM HEPES solution (Sigma), and 20 nM 2-mercaptoethanol (Gibco), under 5% $CO_2$ at 37 °C with a water vapour saturated atmosphere. Cell cultures were maintained at a density of $0.2–1 \times 10^6$ cells/mL.

**ELISA.** THP-1 cells ($9 \times 10^4$) were seeded in flat-bottomed 96-well plates with/without 200 ng/mL LPS (Enzo) and incubated for 2, 4, 6, 12, and 24 h. Following treatment, plates were centrifuged at $130 \times g$ for 5 min at room temperature to pellet cells. Supernatants were transferred to new 96-well plates and stored at −80 °C. ELISA kits for IL-1β (BioLegend, Max Deluxe Set Human), TNFα (Bio-Legend, Max Deluxe Set Human), IL-6 (BioLegend, Max Deluxe Set Human), CXCL10 (R&D Systems, Human DuoSet), and IL-8 (R&D Systems, Human DuoSet) were performed following manufacturer's instructions using Nunc Max-iSorp 96 well flat-bottomed ELISA plates (Thermo Fisher Scientific). Supernatants were prepared in the appropriate reagent diluent using the following dilution factor: IL-1β, neat; TNFα, 1:15; IL-6, 1:5; CXCL10, 1:5; and IL-8, 1:20. ELISA plates were read on a CLARIOstar microplate reader (BMG LABTECH), with cytokine concentrations extrapolated using the MARS Data Analysis Software (BMG LABTECH).

**Immunoblotting.** THP-1 cells ($2 \times 10^6$) were seeded in six-well plates with/without 200 ng/mL LPS and incubated for 2, 4, 6, 12, and 24 h. Cells were lysed in RIPA buffer (150 mM NaCl, 1.0% IGEPAL CA-630, 0.5% sodium deoxycholate, 0.1% SDS, 50 mM Tris, pH 8.0; Sigma) containing $2 \times$ Halt Protease and Phosphatase Inhibitor Cocktail (Thermo Fisher Scientific) for 10 min on ice. Cell lysates were centrifuged at $16,000 \times g$ for 10 min at 4 °C, with supernatants aliquoted into 1.5 mL microcentrifuge tubes and stored at −80 °C. Protein was quantified using the DC Protein Assay Kit II (BioRad), following the manufacturer's instructions. Cell lysates (20 µg/lane) were separated on SDS–PAGE gels and transferred to PVDF membranes (Amersham Hybond P 0.2 µm, GE Healthcare). Membranes were blocked in $1 \times$ Casein Blocking Buffer (Sigma) in phosphate buffered saline-Tween-20 (PBS-T) (137 mM NaCl, 3 mM KCl, 8 mM $Na_2HPO_4$, 1.5 mM $KH_2PO_4$, 0.1% v/v Tween-20) for 1 h at room temperature with gentle agitation and then probed with primary antibodies overnight at 4 °C in blocking buffer with gentle agitation. The blots were then probed with rabbit anti-SQSTM1 (1:2000; Cell Signalling, Cat #: 5114), mouse anti-CLEC11a (1:1000; R&D Systems, Cat #: MAB1904), rabbit anti-cdc42 (1:500; Proteintech, Cat #: 10155-1-AP), rabbit anti-IL-1β (1:5000; Cell Signalling, Cat #: 12703), and rabbit anti-actin (1:5000; Sigma, Cat #: A2066) antibodies overnight at 4 °C in blocking buffer with gentle agitation. Following $3 \times 15$ min washes with PBS-T, membranes were probed with polyclonal goat anti-rabbit IgG-HRP (1:5000;DAKO, Cat #: P0448) or polyclonal goat anti-

mouse HRP (1:5000; DAKO, Cat #: P0447) as appropriate, in blocking buffer for 1 h at room temperature with gentle agitation. After a further $3 \times 15$ min washes with PBS-T, bound antibody was detected using Immobilon Western Chemiluminescent HRP substrate (Millipore), and either exposed to Amersham Hyperfilm ECL (GE Healthcare) or visualised using a myECL Imager (Thermo Fisher Scientific).

**Immunocytochemistry.** THP-1 cells at a density of $8 \times 10^5$ cells/mL were cultured with/without 200 ng/mL LPS and incubated for 2, 4, 6, 12, and 24 h. THP-1 cells were pelleted at $130 \times g$ for 5 min at room temperature followed by one wash with warm PBS containing magnesium and calcium (Gibco). After pelleting as before, cells were fixed in 4% methanol-free formaldehyde in PBS for 20 min at room temperature. Cells were pelleted by centrifugation at $130 \times g$ for 5 min and washed with PBS three times and then resuspended in PBS containing 0.02% sodium azide and stored at 4 °C. Fixed cells ($5 \times 10^4$) were spun onto Plus+ Frost Positive Charged microslides (Solmedia) at 800 rpm for 3 min in a cytocentrifuge (Cytospin 2, Shandon). The attached cells were covered with PBS and circumscribed using a hydrophobic barrier pen (ImmEdge Pen, Vector Labs). Following preincubation with $1 \times$ Perm/Wash solution (BD Biosciences) for 15 min at room temperature, cells fixed with formaldehyde were incubated with 0.01% v/v Triton X-100 in PBS for 5 min followed by $3 \times 5$ min washes with $1 \times$ Perm/Wash solution. Cells were blocked with 10% normal goat serum (NGS, Bio-RAD), 0.1 mg/mL human IgG (Vivaglobin, CSL Behring) in $1 \times$ Perm/Wash solution for 1 h at room temperature. Formaldehyde-fixed cells were incubated with rabbit anti-cdc42 (1:150; Proteintech, Cat #: 10155-1-AP) in 5% NGS in $1 \times$ Perm/Wash solution overnight at 4 °C. Following $3 \times 5$ min washes with $1 \times$ Perm/Wash Solution, cells were incubated with goat anti-Rabbit IgG (H + L) Cross-Adsorbed Secondary Antibody, Alexa Fluor 488® (2 µg/mL; Thermo Fisher Scientific, Cat #: A-11008), and Alexa Fluor® 546 dye-conjugated Phalloidin (2 U/mL; Thermo Fisher Scientific, Cat #: A22283), in 5% NGS in $1 \times$ Perm/ Wash solution for 1 h at room temperature. After $3 \times 5$ min washes with $1 \times$ Perm/Wash solution, cells were incubated with 0.1 µg/mL DAPI in PBS for 15 min at room temperature, followed by a further $3 \times 5$ min washes with $1 \times$ Perm/Wash Solution. Glass coverslips (No 1.5 thickness, VWR) were mounted over stained cells with ProLong Glass Antifade Mountant (Thermo Fisher Scientific). Images were captured using a Leica SP5 confocal microscope at ×63 magnification. Fiji software was used for image processing.

**Cell lysate preparation for time-course analysis.** THP-1 cells ($10^6$) were stimulated with 200 ng/mL LPS (*Salmonella abortus equi* S-form (TLRGRADE™), Enzo) and lysed at appropriate time-points (0, 2, 4, 6, 12, 24 h) in lysis buffer containing 0.2% SDS, 50 mM HEPES pH 7.4 with protease inhibitors (Roche). Benzonase nuclease at a concentration of 25 U/mL (Sigma) was added and incubated for 30 min, on ice. Samples were sonicated for $3 \times 5$ min bursts in a cooled sonicating bath (Bioruptor, Diagenode), to aid solubilisation and then centrifuged at $16,000 \times g$, 10 min, 4 °C to remove insoluble material. The supernatant containing the cell lysate was aliquoted into new tubes and stored at −80 °C. Three biological triplicate time-course experiments were conducted.

**Subcellular fractionation for hyperLOPIT.** Equilibrium density gradient ultracentrifugation is central to the hyperLOPIT protocol and is described in ref. [26]. Briefly, Iodixanol Working solution (IWS) was prepared by dilution of Optiprep™ Density Gradient Medium (60% (w/v) iodixanol, Sigma) to a working concentration of 50% with $6 \times$ lysis buffer (60 mM HEPES pH 7.4, 12 mM EDTA pH 8.0, 12 mM magnesium acetate) containing protease inhibitors. A range of iodixanol-containing solutions were prepared by dilution of the IWS to 6%, 8%, 12%, 16%, 20% and 25% (w/v) iodixanol.

$5 \times 10^8$ THP-1 cells (either unstimulated or following 12 h of 200 ng/mL LPS stimulation) were harvested by centrifugation, washed several times in PBS and resuspended in 20 mL iso-osmotic detergent-free lysis buffer at an approximate concentration of $2.5 \times 10^7$ cells/mL (detergent-free lysis buffer: 0.25 M sucrose, 10 mM HEPES pH 7.4, 2 mM EDTA pH 8.0, 2 mM magnesium acetate, with protease inhibitors). Cell lysis was achieved by passing 1 mL aliquots of the cell suspension through a ball-bearing homogeniser (Isobiotec) with 12 µm clearance, on ice. The efficiency of cell lysis was monitored by Trypan blue exclusion staining and light microscopy. The combined cell lysate was incubated with 25 U/mL

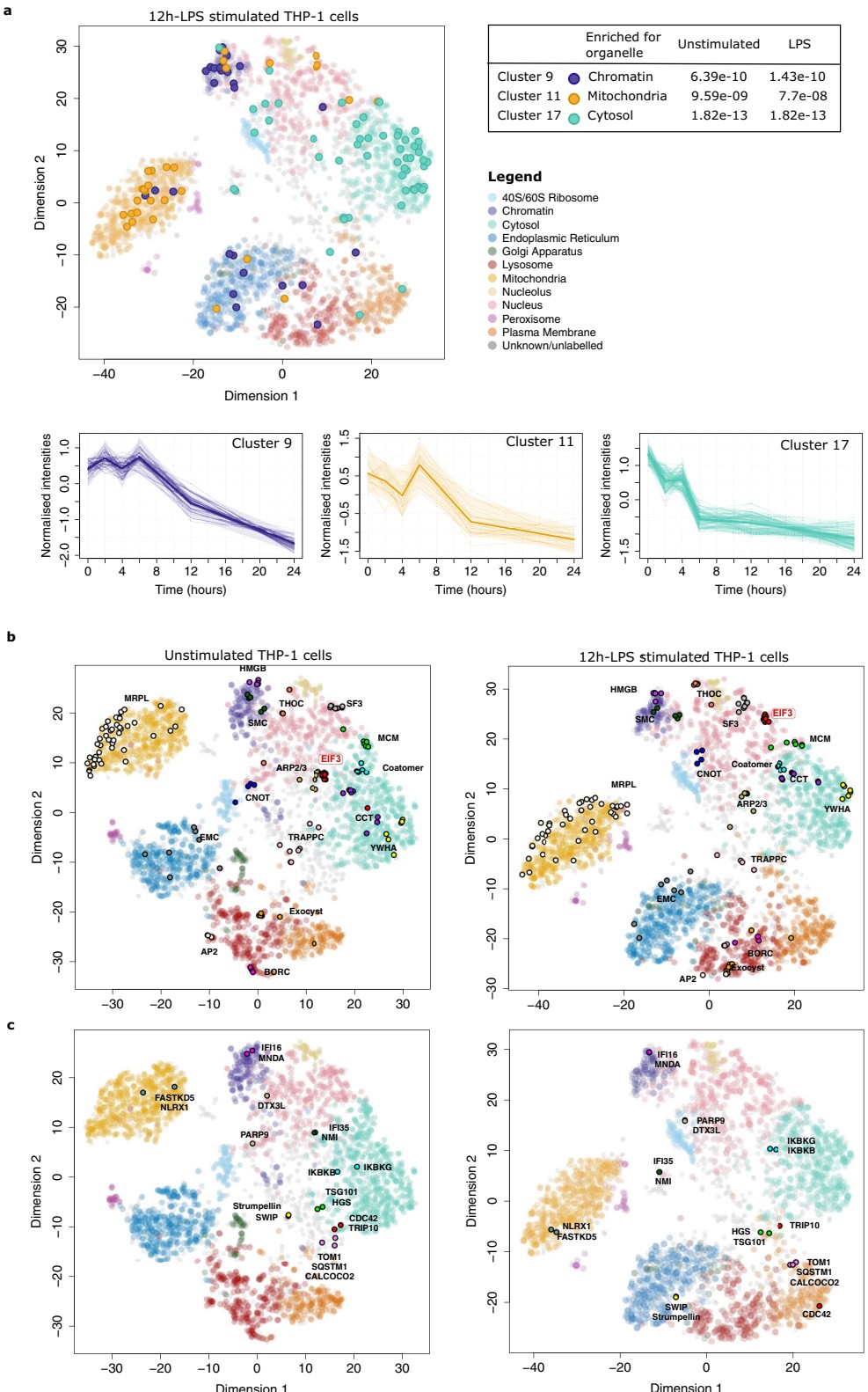

**Fig. 7 The hyperLOPIT dataset can be used as a scaffold to overlay additional layers of spatial information. a** Three temporal clusters from the Bayesian temporal clustering analysis (cluster numbers 9 (purple), 11 (yellow) and 17 (cyan)) were overlaid onto a t-SNE projection of the hyperLOPIT data. A Fisher's exact test was used to determine that these three clusters were enriched for individual hyperLOPIT organelles (*p*-values were adjusted using a Benjamini–Hochberg correction). The temporal profiles of the three Bayesian temporal clusters during the 24 h-LPS time-course are also shown. A single bold line on each profile plot shows the mean normalised MDI profile and the 0.05 and 0.95 quantiles of each cluster is highlighted by shaded bands. **b** Several protein complexes were overlaid onto the hyperLOPIT plot. **c** Protein–protein interaction partner pairs from the literature were shown to co-localise with each other in hyperLOPIT space. Source data are provided as a Source Data file.

Benzonase nuclease (Sigma) for 30 min, on ice and insoluble cell debris was removed by centrifugation at $200 \times g$, 5 min, 4 °C, which was repeated three times.

The lysate solution was underlaid with 6% and 25% (w/v) iodixanol solutions and ultracentrifuged at $100,000 \times g$, 90 min, 4 °C (SW32Ti swinging bucket rotor; Optima L-80 XP Beckman ultracentrifuge). An aliquot of the supernatant taken from above the membrane cushion was diluted 10-fold with chilled acetone for 24 h and then precipitated from the acetone solution at $16,000 \times g$, 30 min, 4 °C, air-dried and stored at −20 °C. The crude membranes from the cushion interface were diluted 5-fold with lysis buffer and ultracentrifuged at $200,000 \times g$, 60 min, 4 °C. Two linear discontinuous gradients were prepared by layering 8%, 12%, 20% and 25% (w/v) iodixanol solutions into 11 mL capacity polyallomer OptiSeal™ ultracentrifuge tubes (Beckman). The gradient was allowed to diffuse for several hours at 4 °C to perform a continuous density gradient, during which time the crude membranes were prepared. The crude membrane pellet was resuspended gently in a final volume of 2.2 mL of 25% iodixanol solution and underlaid beneath the iodixanol density gradient, using a wide-bore needle. Ultracentrifugation was performed at $100,000 \times g$ for 10 h, 4 °C (NVT65 fixed-angle near-vertical rotor; Optima L-80 XP Beckman ultracentrifuge) with minimum deceleration. Subcellular fractions were collected using an Auto-Densi Flow peristaltic pump fraction collector with a meniscus-tracking probe (Labconco). The refractive index of each fraction was determined with a hand-held refractometer (Reichert). Membrane proteins were pelleted from the iodixanol fractions by 3-fold dilution in lysis buffer supplemented with protease inhibitors and ultracentrifugation at $100,000 \times g$, 60 min at 4 °C (TLA-55 rotor; Optima Max-XP benchtop ultracentrifuge, Beckman). This wash step was repeated three times for each fraction, the buffer was removed and the resulting membrane pellets were stored at −80 °C for no longer than one month.

**Proteolytic digestion, TMT labelling and peptide fractionation by UPLC.** All hyperLOPIT fractions were resuspended in urea solubilisation buffer (8 M urea, 0.15% SDS, 25 mM HEPES, pH 8.0), sonicated briefly to aid solubilisation and the protein concentrations were determined using the BCA protein assay (Thermo Fisher Scientific). 50 µg of each membrane sample was labelled with a distinct TMT tag, disulfide bonds were reduced by the addition of 10 mM dithiothreitol (DTT) for 1 h at 37 °C, followed by alkylation of free cysteine residues with 25 mM iodoacetamide (IAA) for 2 h at room temperature in the dark. Each sample was then precipitated overnight with 10 volumes of cold acetone at −20 °C. Samples were centrifuged at $16,000 \times g$, 20 min, 4 °C, the protein pellets were washed with 90% acetone, centrifuged again and finally air-dried for no longer than 15 min. Each sample was resuspended in 0.5 mL of 50 mM HEPES, pH 8.5, sonicated briefly at 4 °C to aid resolubilisation and proteolytically digested with sequencing grade porcine trypsin (Promega) at a final enzyme:protein ratio of 1:20, overnight at 37 °C. Samples were reduced to 0.1 mL in a refrigerated SpeedVac vacuum centrifuge with a cold trap (Labconco).

TMT labelling was conducted as described previously[26]. Amine-reactive 10plex tandem mass tags (TMT, Thermo Fisher Scientific) were equilibrated to room temperature and resuspended in 41 µL LC–MS grade acetonitrile. The peptide-containing samples were added to the appropriate tag vial and labelling was allowed to proceed for 3 h at room temperature on a shaker. The reactions were quenched with 5% (w/v) hydroxylamine solution for 30 min, following which the 10 samples corresponding to each 10plex were pooled together and reduced to dryness by refrigerated vacuum centrifugation. The pooled 10plex sample was then desalted using C18 solid-phase SepPak cartridges (100 mg bed volume, Waters). Twenty fractions including the cytosol-enriched fraction were labelled per gradient, by combining two TMT 10plex experiments in an interleaved labelling design to capture as much subcellular diversity as possible. Therefore, two separate MS runs were conducted for each biological replicate experiment.

The desalted TMT-labelled peptide samples were resuspended in 20 mM ammonium formate, pH 10.0 for high pH reversed-phase UPLC on the Acquity chromatography system with a diode array detector (Waters). Peptides were separated on an Acquity UPLC BEH C18 column (bridged ethyl hybrid C18 column, 2.1 mm i.d. × 150 mm; 1.7 µM particle size, Waters) using acetonitrile (ACN)/ammonium formate (pH 10.0) gradient. MassLynx software (version 4.1) method parameters: 0.0 min: 95% A/5% B; 10.0 min: 95% A/5% B; 60.0 min: 25% A/75% B; 62.0 min: 0% A/100% B; 67.5 min: 0% A/100% B; 67.6 min: 95% A/5% B. Total run time was 75 min, flow rate was set to 0.244 mL/min. Buffer A: 20 mM ammonium formate, pH 10.0; Buffer B: 20 mM ammonium formate, pH 10.0 + 80% (v/v) ACN. Forty fractions were collected across 1 min intervals during peak peptide elution, while monitoring the chromatographic performance with the diode array detector (210–400 nm). The eluted TMT-labelled peptide fractions were dried by vacuum centrifugation and stored until MS analysis. Prior to MS analysis, the 40 fractions were orthogonally combined into 20 fractions by combining fractions which eluted at different points of the UPLC gradient.

For the LPS time-course analysis, 50 µg of each lysate was taken for TMT 6-plex labelling and the volume was made up to 100 µL with 50 mM HEPES, pH 8.0. All reduction/alkylation, tryptic digestion, TMT labelling, sample clean-up and desalting with C18 solid-phase extraction, as well as high pH reversed-phase UPLC were conducted in the same way as described for the hyperLOPIT samples.

**Mass spectrometry.** The LPS time-course data were acquired on an Orbitrap Q-Exactive™ MS (Thermo Fisher Scientific) with MS2 level quantitation, while the

hyperLOPIT analysis was acquired on the Orbitrap™ Fusion™ Lumos™ Tribrid™ instrument (Thermo Fisher Scientific) using synchronous precursor selection (SPS)-MS3 level quantitation. Both instruments were coupled to Dionex Ultimate 3000 RSLCnano systems (Thermo Fisher Scientific).

The TMT-labelled peptide fractions were each resuspended in 0.1% formic acid (FA) and 1 µg of each fraction was loaded onto a micro-precolumn (C18 PepMap 100, 300 µm i.d. × 5 mm, 5 µm particle size, 100 Å pore size, Thermo Fisher Scientific) with 0.1% (v/v) formic acid loading solvent for 3 min, before the valve was switched from load to inject. Peptides were separated on a Proxeon EASY-Spray column (PepMap, RSLC C18, 50 cm × 75 µm i.d., 2 µm particle size, 100 Å pore size, Thermo Fisher Scientific) with a 2–40% (v/v) gradient of ACN + 0.1% FA, at 300 nL/min for 95 min, followed by wash step (70% ACN + 0.1% FA, 5 min) and re-equilibration step, with a total run time of 120 min.

The Orbitrap Q-Exactive™ MS instrument was operated in positive polarity ion mode with data-dependent MS2 (Top 20) acquisition, where the $m/z$ values for precursor ions are measured in the Orbitrap (OT) mass analyser at a resolution of 70,000; AGC target: 1e6; scan range 380–1500 $m/z$; profile mode. Fragment ions were generated by high-energy collisional dissociation (HCD) in the quadrupole mass analyzer and measured in the OT at a resolution of 17,5000; AGC target: 5e4; TopN = 20; isolation window: 1.2 $m/z$; collision energy: 32.5%; stepped collision energy: 10%; profile mode. Peptide ions with charge states of 2+ to 5+ were selected for fragmentation. A total run time (including washing and re-equilibration) of 120 min was employed.

The Orbitrap™ Fusion™ Lumos™ Tribrid™ instrument was operated in positive ion data-dependent mode with an SPS-MS3 acquisition method and a run time of 120 min, as previously described[21,24,26]. The acquisition workflow parameters for XCalibur v3.0.63 (Thermo Fisher Scientific) were set up as follows for SPS-MS3 on the Orbitrap™ Fusion™ Lumos™ Tribrid™ instrument: The full scan was acquired in the OT at a resolution of 120,000; mass range = normal; quadrupole isolation = yes; scan range of 380–1500; RF lens 30%; automatic gain control (AGC) target of 4.0e5; max inject time = 50 ms; microscans = 1, in profile mode with positive polarity; monoisotopic peak determination = peptide; exclude after $n$ times = 1; exclusion duration = 70 s; mass tolerance = 10 ppm; exclude isotopes = yes; include charge state(s): 2–7; perform dependent scan on single charge state pre precursor only = yes; intensity threshold = 5.0e3. MS2 level fragmentation occurred in the linear ion trap (IT) with data-dependent mode = Top Speed (TopS) (ddMS2 IT CID); isolation mode = quadrupole; isolation window of 0.7 $m/z$; activation type = CID (Collision Induced Dissociation); collision energy of 35%, Activation Q = 0.25; Ion trap scan rate = Turbo; AGC target 1.0e4; max inject time = 50 ms; microscans = 1; data type = centroid; precursor selection range = 400–1200; precursor ion exclusion width = low 18 $m/z$, high 5 $m/z$; isobaric tag loss exclusion = TMT; precursor priority = most intense. Synchronous Precursor Selection (SPS) MS3 (ddMS3 OT HCD) was enabled in the Orbitrap using higher collision dissociation (HCD), a number of precursors = 10; MS isolation window = 0.7; HCD Collision Energy of 65%; OT resolution of 60,000; scan range of 100–500 $m/z$; AGC target 5e4; max inject time = 86 ms; microscans = 1; in profile mode. One dataset acquired in August 2016 contained higher AGC target of 1.0e5 and max inject time = 120 ms.

**Data processing.** XCalibur.raw files from both the time-course and hyperLOPIT experiments were processed with Proteome Discoverer v2.1 (Thermo Fisher Scientific) and Mascot server v2.3.02 (Matrix Science). The SwissProt sequence database for *Homo sapiens* was downloaded from UniProt (www.uniprot.org) in November 2016, with 42,118 canonical and isoform sequence entries, together with 48 common contaminants sequences from the cRAP database (common Repository for Adventitious Proteins, https://www.thegpm.org/crap).

For the hyperLOPIT analysis, parameters included a 20 ppm reporter ion integration tolerance window around the expected reporter ion $m/z$ value, 10 ppm precursor mass tolerance, 0.6 Da fragment mass tolerance, reporter peak intensities were integrated using the most confident centroid, tryptic digestion was selected as the enzyme of choice and a maximum of 2 missed cleavages were permitted. Static modifications: TMT(K), TMT(N-term), carbamidomethyl(C); Dynamic modifications: oxidation(M), carbamidomethyl(N-term), carbamyl(K), and deamidation(NQ), as appropriate. Percolator was used to obtain a robust peptide-spectrum match (PSM) level false discovery rate of 1%. Peptide "rank" must equal one with a minimum length of six amino acids, only "high confidence" peptides were used for identification and only unique peptides were used for quantification. Protein Grouping following strict parsimony was enabled and TMT isotopic impurity correction factors were applied. For the time-course analysis, the above parameters were the same, with the exception of the precursor and the fragment mass tolerances which were set to 20 ppm and 0.2 Da, respectively.

For both the hyperLOPIT and the time-course analyses, the PSM-level data was extracted using the following parameters: Number of Protein Groups = 1; Rank = 1; Search Engine Rank = 1; "Isolation Interference" ≤ 50%; "Average Reporter S/N" ≥ 10; "Peptide Quan Usage" = Used; "Quan Info" = Unique. The "Ion Score" was set to ≥20.0 for hyperLOPIT and ≥25.0 for time-course analysis. PSMs corresponding to proteins from the cRAP database, including common laboratory contaminants and preparative agents, such as porcine trypsin and keratins, were removed and all PSMs containing >2 missing values in the reporter ion series were excluded from the analysis. Several PSMs were re-included in the

final hyperLOPIT dataset because they were robustly identified in 11 out of the 12 individual 10plex experiments, where the "missing" 10plex met the following "relaxed" criteria: Isolation Interference < 60%; "Average Reporter S/N" > 9.5; "Search Engine Rank" ≥ 2; "Ion Score" > 19.5. In addition, one TMT channel was removed from the final hyperLOPIT analysis in both the unstimulated and stimulated conditions, due to erroneous labelling of insoluble material during the sample preparation for one replicate. The "Average Reporter S/N" value was recalculated for the nine remaining channels in this 10plex and PSMs with a value less than 9.0 were discarded.

The PSM-level dataset used for the final hyperLOPIT and time-course analyses is openly and freely available in Bioconductor pRolocdata data package (≥v1.27.3)[30]. All data analysis and machine learning were conducted using the R[33](v3.5.1, http://www.R-project.org/) Bioconductor (v3.7)[31,32] packages MSnbase (v2.8.3)[29] and pRoloc (v1.23.2)[30] as per the pRoloc pipeline[26,84].

**Missing value imputation**. A maximum of two missing values per TMT experiment was allowed at the PSM level in both the hyperLOPIT and time-course data. Missing values were carefully assessed in R and values missing at random were imputed with the k-Nearest Neighbour algorithm, and values not missing at random (e.g. biologically relevant missing values such as those resulting from the absence of the low abundance of ions) were imputed with a left-censored deterministic minimal value approach using the MSnbase package[29]. PSMs were quality controlled post-imputation and then combined to protein level by calculating the median of all PSM intensities corresponding to the leading UniProt Accession number for each protein group.

**Data normalisation**. Following missing value imputation, the time-course replicates were combined and normalised using variance stabilising normalisation (vsn)[85] to account for any technical variation between biological replicate experiments without affecting biological variability (Supplementary Data 1). For the hyperLOPIT data, the protein profiles were scaled into the same intensity interval (0 and 1) by dividing each intensity by the sum of the intensities for that quantitative feature (sum normalisation), as described in Mulvey et al.[26] (Supplementary Data 5).

**Data fusion**. 3882 proteins were identified across the three concatenated "unstimulated" hyperLOPIT biological replicates (with a total of 5107, 4838 and 5733 proteins in replicates 1, 2 and 3, respectively), see Supplementary Data 5. 4067 proteins were common across the three "LPS-stimulated" biological replicates (with a total of 4879, 4866 and 5848 in replicates 1, 2 and 3, respectively), see Supplementary Data 5. These combined datasets for the unstimulated and LPS-stimulated experiments were used for downstream data analysis as replicate concatenation has been previously shown to improve dataset resolution[21,86] and the ability to identify genuine residents. These datasets were further subsetted to analyse the proteins common between both conditions (resulting in 3288 proteins) which were used to identify translocation events. The unstimulated and the LPS-stimulated-subsets are found in Supplementary Data 6. All datasets are available in the supporting information and pRolocdata package[30] in Bioconductor.

**Marker list generation**. A list of well annotated, unambiguous resident organelle marker proteins from 11 subcellular niches: mitochondria, ER, Golgi apparatus, lysosome, peroxisome, PM, nucleus, nucleolus, chromatin, ribosome and cytosol, were curated from the UniProt database[87], GO[88] and from mining the literature. Only proteins known to localise to a single location were included as markers. This marker list is available in Supplementary Data 7 and in pRolocdata.

**Protein localisation and re-localisation prediction using a fully Bayesian framework**. A fully Bayesian framework was employed for protein localisation prediction in the R pRoloc package[30] using the TAGM-MCMC algorithm[38], whereby the uncertainty in the allocation of proteins to subcellular compartments was captured and quantified. TAGM-MCMC was used to classify proteins of unknown location to one of the 11 subcellular niches by assessing the output posterior and outlier probabilities for each protein. TAGM-MCMC[38] is a semi-supervised Bayesian generative classifier based on a T-Augmented Gaussian Mixture model (TAGM) that uses Bayesian computation performed using Markov-chain Monte Carlo (MCMC). For each combined dataset, the collapsed Gibbs sampler was run in parallel for 9 chains, with each chain run for 25,000 iterations. The Gelman-Rubin's diagnostic[89] was used to assess the convergence of the 9 Markov-Chains and the 3 best chains were kept and pooled for data processing for each condition. A conservative approach was taken, wherein proteins were only assigned to a subcellular niche if the posterior probability was greater than 0.999 and also if the outlier probability was very small (<1e−6), else proteins were left unassigned and labelled as proteins of "unknown location".

For each protein the predicted localisation was examined in the unstimulated and the LPS-stimulated dataset and relocalisations were mapped and classified into one of four different translocation events: (1) from one organelle class in the unstimulated condition to a different organelle class in the LPS-stimulated dataset i.e. organelle to organelle, (2) from an unknown localisation in the unstimulated dataset to a predicted organelle class in the LPS-stimulated dataset i.e. unknown to

organelle, (3) from a predicted localisation in the unstimulated dataset to an unknown location in the LPS-stimulated dataset i.e. organelle to unknown, and finally (4) a protein that exhibits a large change between posterior probabilities in both conditions and did not get assigned a class label by TAGM.

For every protein the natural L2 distance (also known as the Euclidean norm) was calculated between the TAGM joint posterior probabilities providing an extra source of information on which to rank proteins of interest. A large L2 distance implies a large change in probability distribution. The L2 distance is denoted by

$$d_{L2_{norm}}(x,y) = \sqrt{\sum_{i=1}^{n}(x_i - y_i)^2} \qquad (1)$$

where x and y are the posterior probabilities for the unstimulated and 12 h-LPS stimulated respectively, for each ith class. The L2 distance was used for defining potential type four translocations where proteins did not meet the criteria to be assigned to one of the organelle classes in the training data but did exhibit large changes in their probability distribution as deduced by the L2 value. Only proteins with the maximum L2 distance of 1 were extracted as type 4 candidates for further data mining. This small subset comprised of 30 proteins.

**Assessing changes in abundance**. Limma's paired moderated t-test (using a Benjamini–Hochberg[90,91] (BH) correction, with a 1% FDR) was used to find sets of differentially expressed proteins at each time-point[92]. Proteins were deemed to be significantly up-regulated or down-regulated if both their adjusted p-value ≤ 0.01 and absolute log$_2$FC ≥ 0.6 (Supplementary Data 1).

**Integrative Bayesian model-based clustering with temporal dependencies**. The proteomic expression time-course data were modelled according to a Bayesian mixture model, where the temporal dependencies were captured using Gaussian process (GP) regression models with squared exponential covariance functions[55,93]. Each component in the mixture model may have had different GP hyperparameters, and standard normal priors were placed on the log of each hyperparameter, as in ref. [55]. The number of clusters in the data were automatically inferred[94]. As described by Fritsch and Ickstadt[95], a summarised clustering of the data was produced by first computing the posterior similarity matrix, and then finding the clustering which maximises the posterior expected adjusted Rand index[95]. To assess convergence, two parallel MCMC runs confirmed that the number of clusters in both parallel runs oscillated around the same mode. Convergence was further assessed by monitoring the sampled value of the Dirichlet distribution mass parameter and calculated the Gelman convergence diagnostic[96], which was confirmed to be <1.2 in all three replicates. The association of clusters from the temporal analysis with classifications from the spatial analysis using Fisher's exact test[97] were assessed.

To combine replicate time-series measurements the multiple dataset integration (MDI) Bayesian correlated clustering model[55] was used, which models the dependencies between clustering structures across datasets. Bayesian inference was performed using MCMC (see ref. [55] for further details on inference and modelling) as implemented in the MDI-GPU software[98]. Clusters were extracted only for proteins that were consistently allocated to the same clusterings (sampled from the posterior distribution) across replicates[55,99] (Supplementary Data 3).

**GO enrichment**. Temporal clusters were functionally annotated by using the GO[88]. For each cluster in turn, an enrichment test for biological process, cellular compartment and molecular function annotations, was carried out using the Database for Annotation, Visualisation and Integrated Discovery[100] (DAVID v6.8; https://david.ncifcrf.gov/) where a cutoff of <0.05 was applied according to the Benjamini–Hochberg procedure[90]. The GO annotation enrichment results are found in Supplementary Data 4 and 11.

**R Shiny app**. To enable mining, mapping and visualisation of the data, we provide an interactive R Shiny Application for the community at http://proteome.shinyapps.io/thp-lopit/. The app features two interactive and searchable spatial maps of the two LOPIT experiments for the unstimulated and 12 h-LPS stimulated LOPIT concatenated tri-plicate experiments. The interactive Shiny app in particular provides a platform for the community to interrogate the data directly online via their web browser. The app allows users to visualise the data as annotated t-SNE plots, it supports batch searching, batch import and export of proteins, complexes and networks of interest within the dataset, as well as many other features. The app is subdivided into different tabs: (1) Spatial Map, (2) Profiles, (3) Circos, (4) Table Selection, (5) Table legend, (6) Sample info and (7) Colour picker. A searchable data table containing the experimental feature meta-data is permanently displayed at the bottom of the screen for ease and supports batch import and export of protein information, e.g. accession, gene name, description, etc.

**Reporting summary**. Further information on research design is available in the Nature Research Reporting Summary linked to this article.

## Data availability

The data is freely available online through (1) the R Bioconductor pRolocdata package (≥v1.27.3) (https://bioconductor.org/packages/release/data/experiment/html/

pRolocdata.html), (2) the ProteomeXchange Consortium via the PRIDE (https://www.ebi.ac.uk/pride/) partner repository[101], with the dataset identifier PXD023509 (title: HyperLOPIT and temporal proteomic profiling of the response to lipopolysaccharide in the THP-1 human leukaemia cell line) and (3) an interactive R Shiny app hosted at http://proteome.shinyapps.io/thp-lopit/. Source data are provided with this paper.

## Code availability

As detailed in the data processing section of this manuscript XCalibur.raw files from all proteomic experiments were processed with Proteome Discoverer v2.1 (Thermo Fisher Scientific) and Mascot server (v2.3.02) (Matrix Science). Quantification to protein-level abundances from peptide spectrum matching and subsequent protein localisation analyses was carried out using the freely and openly available R Bioconductor packages MSnbase (v2.8.3) (http://bioconductor.org/packages/release/bioc/html/MSnbase.html), pRoloc (v1.23.2) (http://bioconductor.org/packages/release/bioc/html/pRoloc.html) and pRolocGUI (v2.0.0) (http://bioconductor.org/packages/release/bioc/html/pRolocGUI.html) as described in hyperLOPIT protocols and workflows in[26,85]. All R code used for data analysis and generation of figures is openly and freely available at https://github.com/CambridgeCentreForProteomics/thp-lopit-2021 [102] along with a vignette (https://cambridgecentreforproteomics.github.io/thp-lopit-2021/).

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

## Acknowledgements

The authors would like to thank Dr. Yagnesh Umrania and Julie Howard for their assistance with data analysis and PRIDE submission of proteomics data. This work was supported by Wellcome Trust grants 099135/Z/12/Z and 110071/Z/15/Z. C.M.M. and L.M.B. were supported by a Wellcome Trust Technology Development Grant (grant no. 108467/Z/15/Z). L.M.B. is also supported by EU Horizon 2020 programme INFRAIA project EPIC-XS (Project 823839). A.G. was funded through the Alexander S. Onassis Public Benefit Foundation, the Foundation for Education and European Culture (IPEP) and the Embiricos

Trust Scholarship of Jesus College Cambridge. T.H. was supported by Commonwealth Split Site PhD Scholarship. O.M.C was funded by a Wellcome Trust Mathematical Genomics and Medicine studentship funded by the Cambridge School of Clinical Medicine. D.J.S. was supported by the BRC Oral Health and Disease Theme, NIHR Biomedical Research Centre at UCLH, University College London (BRC727). A.L.R.R. was supported by the CAPES Foundation of the Ministry of Education of Brazil (0698130).

## Author contributions

Author contributions are described according to CRediT standards. Conceptualisation: C.M.M., L.M.B., A.M.S., K.S.L.; Methodology: C.M.M., L.M.B., O.M.C., A.C., M.J.D., A.M.S., K.S.L.; Software and Formal Analysis: L.M.B., O.M.C., L.G.; Investigation: C.M.M., D.J.S., A.L.R.R., A.G., N.K.B., T.H., M.J.D.; Resources: K.S.L. and A.M.S.; Data Curation: C.M.M., L.M.B. and O.M.C.; Writing – Original Draft: C.M.M., L.M.B., O.M.C., A.M.S., K.S.L.; Writing – Review and Editing: C.M.M., L.M.B., O.M.C., D.J.S., A.G., A.C., N.K.B., T.H., A.M.S., K.S.L.; Visualisation: L.M.B., C.M.M., O.M.C.; Supervision and Funding Acquisition: A.M.S. and K.S.L.

## Competing interests

A.C. is an employee of Bristol Myers Squibb. Bristol Myers Squibb had no role in the study design, data collection and analysis, decision to publish, or preparation of the manuscript. All other authors declare no competing interests.
