## [Peer Review File · Nature Communications]

REVIEWER COMMENTS

Reviewer #1 (Remarks to the Author):

The manuscript of Mulvey, et al. reports spatiotemporal proteomic changes during a lipopolysaccharide (LPS)-induced inflammatory response by combination of global proteome analysis with hyperLOPIT in a fully Bayesian framework. This is an interesting work performed with the hyperLOPIT proteomics platform in combination with mass spectrometry, and the technique used in this study is sound, however, there are several mistakes or weak points which should be addressed.

Major points:

1. The authors state that “A total of 311 proteins were found to be altered in abundance during the time course of LPS stimulation, with statistical significance (adjusted p-value < 0.01) ...” (page 7). However, I think the fold change of proteins should also be taken as criteria to determine whether proteins were altered or not. At the same time, in the volcano plot in Figure 2, it is difficult to understand the criteria that the authors used to determine whether the protein is regulated. The judgment criteria of differentially expressed proteins is suggested as the coordinate axis. In Figure 2D, the authors performed Western blot to verify the expression of IL1 β and found that it was increased at all the five time points, and peaked at 4 h after treatment with LPS. However, Figure 2E showed that the expression level of IL1 β was the highest at 24 h after LPS treatment. Could the authors give a reasonable explanation for this result?
2. The authors stated that “only 15 proteins were found to change both their location (at 12 h) and their total protein abundance level in response to LPS over the 24 h time period...” (page 7). What is the biological significance of the 15 proteins selected? Why not to choose the proteins that change both their location and their total protein abundance level over the period of 12 hours for observation?
- 3) When I tried to reproduce the analysis results, I found some potential errors. Are there two columns data, “unstim_rep2_set2_126” and “lps_rep2_set2_131”, missed in the 300804_0_data_set_5371147_qpcbzn.xlsx? If possible, could the authors provide the code to reproduce the analysis and figures in github?

Reviewer #2 (Remarks to the Author):

In this manuscript, Muley et al used the Lilley's lab exciting hyperLOPIT strategy to characterise the spatiotemporal changes in THP-1 cells in response to LPS activation. This work is very interesting as it shows for the first time the changes made to subcellular localisation in response to inflammatory responses. The paper is well structured. The authors provide a web-based tool that allows interrogation of the huge dataset. The tool works very well and is a great way to make the data accessible to readers.

Major comments

1. The proteomics analysis of LPS treated THP1 cells is obviously not that particularly novel. There are many published examples of people doing this for THP1 cells or other macrophages. Nonetheless, the data is required for the spatiotemporal data later in the paper and therefore important.
2. The authors highlight CD14, looking at the map the movement is not totally obvious. In the map, CD14 does still seem to locate closely to other plasma membrane proteins. I understand that PM and endosomes are closely together and not well resolved. But it would be probably better to show the MS profiles of CD14 in supplementary data. It would also be good for the authors to better explain the "moving" of proteins. How do they determine moving "unknown to unknown" organelle.
3. I am intrigued by the translocation of a few proteins. Rab7, for example, which is late endosomal/lysosomal appears to move towards the ER/Golgi, however, not fully. Would you expect this to be a mixed population?
4. Page 21: in order to define the 253 proteins, the authors used a "literature search". Did bioinformatic tools such as GO or Kegg or similar not result in any significant categories?
5. THP1 cells differentiate towards macrophages in response to LPS. It may be good for the authors to discuss that they may also see effects of this.

Minor comments:

1. The authors call the untreated time point 0 hrs. I assume these cells were still grown for 12 hours aside the 12 hr LPS treated cells. I would therefore recommend calling these conditions untreated.
2. Page 9: the 32 IFN regulated proteins are classified as "anti-viral" . as the authors stimulated by LPS they are obviously also anti-bacterial.
3. Figure 2E: how were the secreted cytokines measured? ELISA? Proteomics?
4. Considering the accuracy of the approach, I don't think the values in the supplementary tables require 8 digits.

5. There is a graph in supplementary table 1 (I guess at least it is that one, the system gives terrible names ending with 133_qpcbzn). I am pretty sure this graph should not be there.
6. It would be nice having the protein or at least the gene names in supplementary tables 9 (in the files ending with 149... and 148...
7. Page 25: SCARB1 is not a transporter, but a receptor.
8. Page 27: to my knowledge CHMP2b and Ist1 are both in the ESCRT II complex. It may be interesting to look at the whole complex.

Matthias Trost

Reviewer #3 (Remarks to the Author):

The manuscript combines spatial proteomics with LPS stimulation, to show how proteins can end up at different locations after a stimulation. It is a very interesting combination of approaches, and the data look very convincing. In principle I am all in favor of publishing, but there are some major issues that have to be dealt with first.

With all due respect to the authors, in general the manuscript breathes a bit “running a black box computational tool on the data without a thorough analysis” of the tool or the results and with some weird observations about the biology.

The experiment has been done in threefold, but the results presented in the main text only discuss the proteins consistently classified across the three. How much variation is there in the spatial classification if you e.g. do the classification only on the proteins detected in all three replicates and then separately for the three replicates. The manuscript reports confident localizations for about

half of the proteins detected in all three replicates. What happened to the other ones? Were they localized to multiple organelles? Or not confidently to the same organelle? Have you done cross validation of the localizations by leaving out marker proteins? There is actually very little detail given about the data processing. The L2 that is mentioned a number of times is never introduced. I doubt whether the nature communications readership knows what it is (I had to look it up), nor is explained why the L2 distance would be the best distance to determine relocation.

The "unknowns" in Figure 4 and in the text. To what extent could they not have moved, but just did not have high enough posterior probability scores? In other words, if we do not know where they have ended up we cannot argue that they have moved between organelles. From the methods I understand that "unknown location" is used for proteins that could not unambiguously be assigned to a location. We cannot argue proteins moved if we cannot confidently assign them to a location in the cell. Unless I am missing something, I do find this a rather big oversight. Why is the "unknown location" not depicted in Figure 4? Is it "all over the place"? Then I find results regarding the relocation to an unknown location not worthwhile. The manuscript & figures would gain from not discussing those and focusing on what can be confidently be concluded.

The manuscript mentions "Proteins were ranked according to highest L2 distance and a stringent threshold was applied (posterior probability > 0.999; outlier probability < 1e-6)" I understand from other parts of the text that this posterior probability refers to the specific location, not to the relocation, or does it? The authors furthermore write "events were filtered for the natural L2 distance between TAGM joint posterior probabilities." But later only mention a ranking, not an explicit filtering at e.g. some cutoff value.

The manuscript writes about proteins moving from one location to another, but to what extent are the proteins actually moving, or are newly produced proteins targeted to different locations (and the old ones degraded)? Proteins actually leaving the mitochondria (the authors talk about "export" here) is associated with apoptosis, so I guess not all cases of proteins being in different locations after LPS are cases of actual relocalizations. Twelve hours is enough time for transcription/translation to take place after an LPS stimulus as the authors show by finding proteins with different overall concentrations (the lack of a difference for any specific protein is IMHO not enough of an argument that it actually moved)

Supplement Figure 2a: if the authors stick to the proteins that move between known locations (leave out the unknowns), they might be able to use arrows or so, to make a more convincing case about relocalizations.

"Three clusters from the Bayesian temporal clustering analysis" why these three?

Where does the discrepancy between figures 4A and supplementary Figure 1G come from? The numbers are very different.

Regarding “Protein-protein interaction partners are also preserved during the hyperLOPIT protocol (Figure 7C). For example, we see co-localisation of (i) CDC42 and TRIP10 (CDC42-interacting protein 4)73” In 7C the proteins are close together before stimulation, but not anymore after stimulation, so we are only supposed to look at the left panel? With respect to NLRX1 and FASTKD5. I only see one dot? Are they on top of each other? If so, that might have been mentioned explicitly.

I am not convinced by the added value of the information of supplemental figure 5. This appears rather superficial comparing of proteomics data, without a specific conclusion.

Smaller things that deserve some explanation: Why would “cellular respiration” be reduced after LPS?

and

“CD14, the classic co-receptor for LPS, was localised to the PM in unstimulated conditions, however was no longer associated with the PM following stimulation”

so where did it go?

Editorial: check the use of the words “which” vs. “that”

Please do not use “major” and “key” to make things sound more important.

Editorial: “This data indicates that distinct relocalisation events and abundance changes are both necessary to drive the global, cellular response to LPS.” You cannot state that without testing it experimentally, furthermore. Rephrasing to “are part of the response to LPS” sounds more logical given the data.

Professor Kathryn Lilley, FRSB

Department of Biochemistry
Cambridge Centre for Proteomics,
Department of Biochemistry
University of Cambridge
Tennis Court Road
Cambridge
CB2 1PD
Tel: +44(0)1223 760255
E-mail: k.s.lilley@bioc.cam.ac.uk

We thank all the reviewers for their thoughtful and thorough comments on our manuscript. We are pleased that all three reviewers are enthusiastic about the work and its importance, and, in principle, are keen to see the work published.

Reviewer #1 (Remarks to the Author):

The manuscript of Mulvey, et al. reports spatiotemporal proteomic changes during a lipopolysaccharide (LPS)-induced inflammatory response by combination of global proteome analysis with hyperLOPIT in a fully Bayesian framework. This is an interesting work performed with the hyperLOPIT proteomics platform in combination with mass spectrometry, and the technique used in this study is sound, however, there are several mistakes or weak points which should be addressed.

Major points:

1. The authors state that “A total of 311 proteins were found to be altered in abundance during the time course of LPS stimulation, with statistical significance (adjusted p-value < 0.01) ...” (page 7). However, I think the fold change of proteins should also be taken as criteria to determine whether proteins were altered or not. At the same time, in the volcano plot in Figure 2, it is difficult to understand the criteria that the authors used to determine whether the protein is regulated. The judgment criteria of differentially expressed proteins is suggested as the coordinate axis. In Figure 2D, the authors performed Western blot to verify the expression of IL1 β and found that it was increased at all the five time points, and peaked at 4 h after treatment with LPS. However, Figure 2E showed that the expression level of IL1 β was the highest at 24 h after LPS treatment. Could the authors give a reasonable explanation for this result?

We thank the reviewer for bringing this to our attention and apologise for mistakenly omitting the fold change criteria from the manuscript. To determine whether proteins were changed in abundance during the LPS time-course we used the R limma paired moderated t-test (using a Benjamini-Hochberg correction) and required the adjusted p-value ≤ 0.01 and a minimum absolute threshold of $\log_2FC \geq 0.6$, to find sets of differentially expressed proteins at each time-point. We have

added the fold change criteria to the manuscript in the methods section and have re-generated the volcano plots in Figure 2 so they include dotted lines on each plot to indicate the fold change cutoff and raw p-value cutoff (where it is equivalent to the adjusted p-values on the x-axis). We have also updated all the legends in Figure 2 for clarity.

The variation in the peak of protein expression for IL-1 β could result from the different ways IL-1 β is detected with the three experimental techniques used. The antibody for western blotting (Cell Signalling #12703) only recognises the full-length IL-1 β (31KDa) and not the mature form (17 KDa), see Fig 2D. Mass spectrometry (MS) however (Fig 2E red line), identifies peptides from both forms irrespective of processing. The MS peptide-level results have identified peptides from the entire protein sequence (see IL-1 β protein sequence below), indicating that the MS data for this protein cannot differentiate between the active mature form and the full-length pro-form. The ELISA results (Fig 2E blue line) clearly demonstrate that mature IL-1 β is still being secreted between 12 to 24 hours post LPS, so we are confident that IL-1 β is being produced at high levels between these two timepoints. Our results would suggest that the majority of the intracellular IL-1 β is being converted to the mature form at the 12 and 24 hour time points and then secreted. Hence, we observe the western blot bands at 12 and 24 hour time points being diminished, the intracellular MS proteomic abundance levels remaining high and the secreted levels rapidly rising between these two timepoints, as seen by ELISA.

10	20	30	40	50
MAEVP	ELASE	MMAYYS	GNED	DLFFEADGPK
				QMKCS
				SFQDLD
				LCPLDGGIQL
60	70	80	90	100
RISDH	HYSKG	FRQAASV	VVA	MDKLRKMLVP
				CPQTFQENDL
				STFFPFIFEE
110	120	130	140	150
EPIFF	DTWDN	EAYVHD	APVR	SLNCTLRDSQ
				QKSLVMSGPY
				ELKALHLQGG
160	170	180	190	200
				DMEQQVVFMS
				SFVQGEESND
				KIPVALGLKE
				KNLYLSCVLK
				DDKPTLQLES
210	220	230	240	250
				VDPKNYPKKK
				MEKRFVFNKI
				EINNKLEFES
				AQFPNWIYST
				SQAENMPVFL
260				
				GGTKGGQDIT
				DFTMQFVSS

Full length protein sequence of IL-1 β (Uniprot ID P01584). The biologically active cleaved mature form is highlighted in yellow (amino acids 117 – 269). The peptides identified in the time-course study may correspond to the full-length pro-form and/or the active cleaved form (shown in red).

2. The authors stated that “only 15 proteins were found to change both their location (at 12 h) and their total protein abundance level in response to LPS over the 24 h time period...” (page 7). What is the biological significance of the 15 proteins selected? Why not to choose the proteins that change both their location and their total protein abundance level over the period of 12 hours for observation?

Gene ontology analysis of the 15 proteins revealed no significant GOBP terms linked to this small group of proteins. Only two proteins were found to alter in both abundance and location specifically at the 12 h time-point, SQSTM1 (Q13501) and CLEC11A (Q9Y240). We therefore chose these two proteins for further observation with western blotting, which confirmed the MS results (Fig 6C and 6D). We did attempt to generate immunofluorescence images for SQSTM1 and CLEC11A, but the

antibodies were found to not work under any of the conditions we tried. We also discussed the relevance of these two proteins to the LPS response in the Results text (see Pages 8, 18, 21 of the manuscript).

We intentionally did not use the usual experimental technique of transfection and over-expression of reporter constructs as a means to verify localization. This was for two main reasons. Firstly, the strength of this study is its focus on native proteins and their dynamics without the interference of unregulated expression and the issues associated with reporter tags, such as dimerization, steric interference and alternative potential protein interaction. Secondly, transfection, either using lipid mediators or viral particles, of THP-1 cells and more broadly innate immune cells, results in cellular activation and a major switch in cellular phenotype, making comparisons between experiments highly problematic.

As there was no common biological process or pathway shared between the other 13 proteins that were found to change location and abundance at 24 h, we made the decision to focus our resources, time and efforts on other aspects of the paper.

3) When I tried to reproduce the analysis results, I found some potential errors. Are there two columns data, "unstim_rep2_set2_126" and "lps_rep2_set2_131", missed in the 300804_0_data_set_5371147_qpcbzn.xlsx? If possible, could the authors provide the code to reproduce the analysis and figures in github?

We thank the reviewer for this comment and have added a comprehensive open-source freely available step-by-step R vignette on Github, to allow readers to reproduce for the data analysis and figures in this manuscript and also apply it to their own data.

The HTML vignette is available to view at <https://cambridgecentreforproteomics.github.io/thp-lopit-2021> and the associated Github repository is located at <https://github.com/CambridgeCentreForProteomics/thp-lopit-2021>. It has been referenced in the Supplementary Information on page 10 and Code Availability section on page 42 of the main manuscript .

As stated in the Data Processing section of the Supplementary Information (page 14 of SI) the TMT 126 channel in replicate 2 of the unstimulated data (which corresponds to fraction 27) was removed from the final hyperLOPIT analysis and the equivalent channel in the LPS stimulated LOPIT dataset (TMT 131 channel in replicate 2, fraction 27) was also removed, due to erroneous labelling of insoluble material during the sample preparation. The "Average Reporter S/N" value was recalculated for the nine remaining channels in replicate 2 of this 10plex and PSMs with a value less than 9.0 were discarded in Proteome Discoverer before dataset concatenation. For the convenience and clarity of readers, we have added a note stating a total of 59 channels were used for data analysis in each condition, to the "Column legends" tabs of the spreadsheets named Supplementary Table 5 and 6, which contain the protein quantitation data.

Reviewer #2 (Remarks to the Author):

In this manuscript, Mulvey et al used the Lilley's lab exciting hyperLOPIT strategy to characterise the spatiotemporal changes in THP-1 cells in response to LPS activation. This work is very interesting as it shows for the first time the changes made to subcellular localisation in response to inflammatory responses. The paper is well structured. The authors provide a web-based tool that allows interrogation of the huge dataset. The tool works very well and is a great way to make the data accessible to readers.

Major comments

1. The proteomics analysis of LPS treated THP1 cells is obviously not that particularly novel. There are many published examples of people doing this for THP1 cells or other macrophages. Nonetheless, the data is required for the spatiotemporal data later in the paper and therefore important.

Throughout the manuscript we have tried to acknowledge the fact that other research groups have utilised the THP1 cell line for proteomics analysis of LPS stimulation and we have cited their research accordingly, please see Introduction (page 5), Discussion (page 24) and Supplementary Tables 9 and 10. We believe the novelty of our work lies in (1) the global approach taken to analyse the whole cellular response to LPS stimulation, rather than to focus on a specific subset of proteins or a particular location of interest, and (2) the Bayesian analysis for quantifying relocalisation events in response to LPS. We hope that our work will be of major interest to the other research groups interested in LPS signalling and that the shiny app provides an easy to use resource for the research community. We thank the Reviewer for recognising the importance of including the initial proteomics analysis for interpreting the spatiotemporal data presented in this study.

2. The authors highlight CD14, looking at the map the movement is not totally obvious. In the map, CD14 does still seem to locate closely to other plasma membrane proteins. I understand that PM and endosomes are closely together and not well resolved. But it would be probably better to show the MS profiles of CD14 in supplementary data.

CD14 was not identified in the list of "movers", rather it was localised to the plasma membrane in the unstimulated condition and remained unclassified in the LPS-stimulated condition, suggesting that it is no longer tightly associated with the plasma membrane cluster by 12 h of LPS. This protein was not identified as undergoing relocalisation in this analysis, as it did not reach the stringent criteria we used for defining a translocation event. Despite this, we highlighted CD14 in the Results section as we felt the readers would be interested in a protein which is known to be critical for TLR4- driven TRIF-mediated IFN expression upon LPS stimulation, a process which requires CD14/TLR4 complex endocytosis.

CD14 was found to be upregulated in abundance in response to LPS at 24 hours (mentioned on page 9), while the LOPIT data indicates that its strong association with the plasma membrane is lost after LPS stimulation. The well characterised CD14/TLR4 mediated endocytosis contributes to the abundant upregulation of IFN response genes, a signature which we observed in the time-course dataset. To address the Reviewer's comment, we have now updated the text to clarify that this protein was not identified as a "mover" in the relocalisation analysis on page 14, and we have included the MS profiles in the supplementary information (Supplementary Figure 1H), as requested.

It would also be good for the authors to better explain the "moving" of proteins. How do they determine moving "unknown to unknown" organelle.

We have updated the main text on page 15 and methods on page 39 to clarify the moving of proteins

and in particular how type 4 movers (unknown-to-unknown) were extracted. Type 4 relocalisations are proteins that did not get confidently assigned to one of the subcellular classes in the training data by TAGM (they were left as unknown/unassigned), but did exhibit large changes in their probability distributions following LPS stimulation.

The unknown proteins are not an organelle class but an umbrella term we use to describe proteins that did not get classified to one of the eleven organelles found in the training set for TAGM-MCMC, which generally arise from one of several scenarios. These scenarios are (i) the proteins steady state is between multiple organelles, (ii) the protein is located to a subcellular niche not described in the training data, (iii) the protein did not meet the strict classifier cutoff for localising to the organelle for which it is assigned.

In order to extract type 4 relocalisations the change in probability distribution was measured by calculating the L2 distance between the probability distributions in each condition (see below). Proteins with a maximum L2 distance of 1 were considered type 4 relocations. We found 30 proteins that met this criteria.

page 15 has been updated to clarify types 4 movers -

Based on the combination of joint posterior and outlier probabilities, protein trans/relocalisation events that occur following LPS stimulation were classified as four distinct scenarios: (i) Type 1: organelle-to-organelle, (ii) Type 2: unknown-to-organelle, (iii) Type 3: organelle-to-unknown, and (iv) Type 4: unclassified proteins that exhibit large changes in their posterior probability distribution between conditions. For additional stringency and to capture large movements in probability space, all potential relocalisation events were ranked according to their protein's natural L2 distance between their TAGM joint posterior probabilities. A large L2 distance between probabilities is indicative of proteins that exhibit large movements in probability space.

We have also updated the methods on page 39 to further clarify how the L2 distance was calculated and used in extracting proteins that had changes in their probability distribution following LPS.

page 39 has been changed to include -

For every protein the natural L2 distance (also known as the Euclidean norm) was calculated between the TAGM joint posterior probabilities, providing an extra source of information on which to rank proteins of interest. A large L2 distance implies a large change in probability distribution. The L2 distance is denoted by

$$d_{L2norm}(x, y) = \sqrt{\sum_{i=1}^n (x_i - y_i)^2}$$

where x and y are the posterior probabilities for the unstimulated and 12h-LPS stimulated respectively, for each i th class. The L2 distance was also used for defining potential type 4 translocations where proteins were left as "unknown" and did not meet the criteria to be assigned to one of the organelle classes in the training data. Proteins with the maximum L2 distance of 1 were extracted as type 4 for further data mining. This small subset comprised of 30 proteins.

3. I am intrigued by the translocation of a few proteins. Rab7, for example, which is late endosomal/lysosomal appears to move towards the ER/Golgi, however, not fully. Would you expect this to be a mixed population?

We thank the Reviewer for raising this interesting point. RAB7A was identified as a Type 1 translocation event and was found to relocate from the lysosome to the ER following LPS stimulation. The TAGM-MCMC posterior probability distribution for RAB7A (see the uncertainty violin plot shown below) shows that in unstimulated conditions RAB7A is localised to the lysosomal fraction with high probability but interestingly also has a very low probability of being ER-associated. After 12h-LPS stimulation we find that RAB7A has entirely relocated to the ER. This observation is supported by the literature, which shows that RAB7A modulates ER stress and ER morphology (Mateus et al., 2018, Biochim Biophys Acta Mol Cell Res, doi: 10.1016/j.bbamcr.2018.02.011). It is known that LPS stimulation induces ER stress via TLR4 (Woo et al., 2009, Nat Cell Biol, doi: 10.1038/ncb1996), which may account for the relocalisation of RAB7A. From the LOPIT data we do not have reason to believe RAB7A exists as a mixed population, in this instance. It is important to be cautious when interpreting multivariate localisation data based on the apparent position of a protein in a two-dimensional t-SNE plot. The Bayesian analysis however captures uncertainty in the distribution of proteins between organelles and is therefore able to investigate dynamic redistribution of native proteins in LOPIT datasets as shown below.

Uncertainty violin plot for localisation of RAB7A in the unstimulated and in LPS-stimulated datasets.

4. Page 21: in order to define the 253 proteins, the authors used a “literature search”. Did bioinformatic tools such as GO or Kegg or similar not result in any significant categories?

The 253 proteins which were identified as undergoing relocalisation in this study represent many

diverse and dynamic functions localised across different subcellular regions. In order to ensure that we identified as many biologically relevant processes as possible, each protein was manually curated by a literature search (see Supplementary Table 9). We opted for this approach as we felt a GO annotation enrichment analysis alone of 253 individual proteins would yield too many broadly defined biological terms which would be impractical to discuss within the limitations of the manuscript.

In response to the Reviewer's comment, we have now conducted a GO annotation enrichment analysis of the 253 proteins which is included as Supplementary Table 11 and is described on page 15. This analysis revealed enrichment of 133 distinct GPBP terms (Benjamini corrected p-value < 0.05), including the top-scoring term "GO:0001731~formation of translation preinitiation complex" (Benjamini p-value 2.20E-11). Interestingly, three of the most significantly enriched terms were "GO:0008104~protein localization" (Benjamini p-value 2.52E-09) containing 76 proteins from the set of 253, "GO:0045184~establishment of protein localization" (Benjamini p-value 3.28E-09) with 67 proteins, and "GO:0015031~protein transport" (Benjamini p-value 5.55E-09) with 63 proteins. This analysis confirmed that protein trafficking and translocation are important functions for this group of relocalising proteins.

5. THP1 cells differentiate towards macrophages in response to LPS. It may be good for the authors to discuss that they may also see effects of this.

Monocyte-to-macrophage-like differentiation of THP-1 cells occurs in response to LPS and we have observed that this process is underway by 12h of LPS stimulation in our study. In response to the Reviewer's comment, we have now added additional sentences to pages 9 - 11 to emphasise this point. We have also highlighted several proteins whose altered abundance corroborates this transition to a more macrophage-like phenotype and have provided supporting citations. For example, downregulation of RET (Nakayama, S. *et al. Br J Haematol* 105, 50-57 (1999)), SRGN (Chang, M. Y. *et al. J Biol Chem* 287, 14122-14135, doi:10.1074/jbc.M111.324988 (2012)) and ITGAX, as well as upregulation of EPST11 (Kim, Y. H., Lee, J. R. & Hahn, M. J. *Biochem Biophys Res Commun* 496, 778-783, doi:10.1016/j.bbrc.2017.12.014 (2018)), PLEKHO2 (Zhang, P. *et al. Cell Signal* 37, 115-122, doi:10.1016/j.cellsig.2017.06.006 (2017).) and ICAM1 together support the transition to a macrophage-like state. We hope that the observation of macrophage polarisation is now more clearly addressed in the text.

Minor comments:

1. The authors call the untreated time point 0 hrs. I assume these cells were still grown for 12 hours aside the 12 hr LPS treated cells. I would therefore recommend calling these conditions untreated.

This has been addressed in the text and changed to untreated.

2. Page 9: the 32 IFN regulated proteins are classified as "anti-viral" . as the authors stimulated by LPS they are obviously also anti-bacterial.

Anti-viral has been changed to "anti-microbial" to embrace both anti-viral and anti-bacterial responses. The unique ability of TLR4 to activate the classical termed anti-bacterial MYD88 and anti viral TRIF signalling cascades, means that the use of the phrase anti-microbial is probably more

accurate. We thank the reviewer for highlighting this point.

3. Figure 2E: how were the secreted cytokines measured? ELISA? Proteomics?

The secreted cytokines were measured by ELISA, as described in Supplementary Methods. The text and the figure legend have now been updated to clarify this.

4. Considering the accuracy of the approach, I don't think the values in the supplementary tables require 8 digits.

All numerical values in the supplementary tables have been updated and rounded to 4 significant figures.

5. There is a graph in supplementary table 1 (I guess at least it is that one, the system gives terrible names ending with 133_qpcbzn). I am pretty sure this graph should not be there.

Thank you, this graph has now been removed

6. It would be nice having the protein or at least the gene names in supplementary tables 9 (in the files ending with 149... and 148...

The accession numbers for each protein and their corresponding gene name is listed in Supplementary Table 9

7. Page 25: SCARB1 is not a transporter, but a receptor.

The word transporter has now been removed.

8. Page 27: to my knowledge CHMP2b and Ist1 are both in the ESCRT II complex. It may be interesting to look at the whole complex.

We thank the reviewer for pointing out this complex to us. The Endosomal Sorting Complexes Required for Transport ESCRT-III complex is involved in the formation of multivesicular bodies (MVBs) and endosomal sorting. Indeed, we can identify eleven components of the ESCRT-III machinery and have now included these as a supplementary figure (Supplementary Figure 4C) and mention them briefly in the text (see page 20). Interestingly, six of these proteins become closely associated in the stimulated condition (CHMP1A, CHMP1B, SPAST, CHMP2A, CHMP2B, VPS4A). Of these, CHMP2B was identified as a Type 2 mover (unknown to lysosome), whereas CHMP1A, CHMP1B and SPAST were also classified as unknown in unstimulated condition and lysosomal following LPS-stimulation, however they did not meet the threshold criteria to qualify as Type 2 movers. The co-localisation of these proteins from an important signalling complex further demonstrates the strength of the LOPIT approach for investigating protein complexes.

Matthias Trost

Reviewer #3 (Remarks to the Author):

The manuscript combines spatial proteomics with LPS stimulation, to show how proteins can end up at different locations after a stimulation. It is a very interesting combination of approaches, and the data look very convincing. In principle I am all in favor of publishing, but there are some major issues that have to be dealt with first.

With all due respect to the authors, in general the manuscript breathes a bit “running a black box computational tool on the data without a thorough analysis” of the tool or the results and with some weird observations about the biology.

The experiment has been done in threefold, but the results presented in the main text only discuss the proteins consistently classified across the three. How much variation is there in the spatial classification if you e.g. do the classification only on the proteins detected in all three replicates and then separately for the three replicates. The manuscript reports confident localizations for about half of the proteins detected in all three replicates. What happened to the other ones? Were they localized to multiple organelles? Or not confidently to the same organelle? Have you done cross validation of the localizations by leaving out marker proteins? There is actually very little detail given about the data processing. The L2 that is mentioned a number of times is never introduced. I doubt whether the nature communications readership knows what it is (I had to look it up), nor is explained why the L2 distance would be the best distance to determine relocation.

We thank the reviewer for their opinion on the data analysis and apologise for the lack of clarity. Several of the authors are R Bioconductor developers and we pride ourselves on producing open source open-development software. In response to these comments we have written a comprehensive R vignette (tutorial) to accompany the manuscript with all source code and step-by-step details on how to reproduce the data analysis and figures so that others can repeat the workflow and use the methods in their own work. The vignette is located as at <https://cambridgecentreforproteomics.github.io/thp-lopit-2021> and the associated Github repository is located at <https://github.com/CambridgeCentreForProteomics/thp-lopit-2021>. containing all code needed to reproduce the analysis and generate the figures. The vignette largely follows the already published pRoloc pipeline for spatial proteomics data analysis which is openly available in Bioconductor (and has also been published in Breckels LM, et al. F1000Research 2018, 5:2926 and Mulvey, C., Breckels, et al. Nat Protoc 12, 2017, 1110–1135). We have addressed the more specific queries from the reviewer below.

With regards to upstream data analysis, we follow the pRoloc pipeline for both the processing of the LOPIT and shotgun data (as mentioned above) where we use dataset concatenation (also called dataset fusion) prior to classification, rather than conducting 3 separate machine learning analyses and using a voting scheme across the classifications results. We chose to use data fusion as this not only has been widely validated and assessed in several independent papers (e.g. Trotter et al. Proteomics. 2010, Dec;10(23):4213-9, Christoforou, A. et al. Nat. Commun. 2016, 7:8992, Geladaki, A., et al. Nat Commun. 2019, 10, 331, among others), it has also been shown to improve classifier accuracy and most importantly it avoids any experiment specific bias that could arise if the latter approach was used and analysed on a replicate by replicate basis.

We report confident localisations for 2,496 and 2,500 proteins in the unstimulated and 12h-LPS stimulated respectively (1,713 and 1,717, not including the 783 marker proteins). The remaining unclassified proteins (referred to as “unknown” or “unlabelled” throughout this manuscript) in each condition are 792 and 788 proteins for unstimulated and 12h-LPS stimulated, respectively. These

proteins did not confidently get assigned to one of the 11 subcellular organelle classes described in the training data. These proteins are highlighted as grey points in all t-SNE maps in the manuscript. We apologise for missing this from the figure legends of the t-SNE plots and thank the reviewer for bringing this to our attention. We have now modified the legend in all t-SNE plots to include this. It would be interesting to follow up on the unknown proteins in detail but it is beyond the scope of the manuscript. Proteins that are labelled unknown/unlabelled generally fall into one or more of three categories: (i) the proteins steady state is between multiple organelles, (ii) the protein is located to a subcellular niche not described in the training data, (iii) the protein did not meet the strict classifier cutoff for localising to the organelle for which it is assigned. The resulting data can be found for readers in an online interactive R Shiny application (<http://proteome.shinyapps.io/thp-lopit/>) which is freely available for the community to interrogate and follow-up their own proteins of interest.

Cross-validation is widely used to estimate frequentist uncertainty. The analysis here already incorporates the uncertainty due to variations across replicates (in the Bayesian clustering) marker choices using the TAGM Bayesian model, which is a cleaner approach.

We apologise for not explicitly including the L2 distance metric (also known as the L2 norm) and thank the reviewer for highlighting this point. We have modified the manuscript to include this in the methods on page 39. We chose the L2 norm as our distance metric as it calculates the distance of the vector coordinate from the origin of the vector space, here this is probability space and the result is a positive distance value which is straightforward to interpret. For example, a protein allocated to two different organelles would have distance 1 e.g. EIF3F has a probability of 1 for the Cytosol and 0 for all other organelle (see below and Supplementary Table 6 of LOPIT results) in the unstimulated, and a probability of 1 for Nucleus and 0 for all other compartments in LPS. Thus a protein allocated to the same organelle would have distance 0. Organelles that shift, for example if a protein is split 0.25, 0.75 between two organelles moved to 0.75 0.25 this would have a distance $\sqrt{(0.5^2 + 0.5^2)}$ approx 0.71. We have provided a few examples below.

The below violin plots show the TAGM-MCMC posterior probability distribution for O00303 (EIF3F). It is assigned to the cytosol with membership probability of 1 in unstimulated and probability of 1 to the nucleus in LPS. Its L2 probability difference is thus 1. This protein has been classed as a type 1 translocation in these data.

Distribution of subcellular membership for O00303 (EIF3F)

Uncertainty violin plot for localisation of EIF3F in the unstimulated and in LPS-stimulated datasets.

The violin plot below shows the membership probability distribution of the protein O75689 (ADAP1) is assigned unknown in LPS as its distribution is split over several organelles and its assigned plasma membrane in LPS with a probability of 1. This protein is classed as a type 3 mover with an L2 distance of 0.537

Distribution of subcellular membership for O75689 (ADAP1)

Uncertainty violin plot for localisation of ADAP1 in the unstimulated and in LPS-stimulated datasets.

The "unknowns" in Figure 4 and in the text. To what extent could they not have moved, but just did not have high enough posterior probability scores? In other words, if we do not know where they have ended up we cannot argue that they have moved between organelles. From the methods I understand that "unknown location" is used for proteins that could not unambiguously be assigned to a location. We cannot argue proteins moved if we cannot confidently assign them to a location in the cell. Unless I am missing something, I do find this a rather big oversight. Why is the "unknown location" not depicted in Figure 4? Is it "all over the place"? Then I find results regarding the relocation to an unknown location not worthwhile. The manuscript & figures would gain from not discussing those and focusing on what can be confidently be concluded.

The "unknowns" are not a subcellular class, they are the name we give to proteins which were not classified to one of the organelles in the training data. They are indeed proteins which did not meet the classifier threshold to allow them to be confidently assigned to one of the 11 subcellular classes in the training data. This is mentioned at the top of pages 14 and 39 of the Methods. For translocation events 2, 3 and 4 (the cases where we report a movement to/from or involving an "unknown" assignment), we do not claim that proteins translocate between discrete classes. From examining the change in their classifier label (as output from TAGM) coupled with their change in posterior distribution from TAGM-MCMC (as deduced by the L2 distance) we know there is a change

in their profile signal between the two conditions but we can not conclude for the “unknown” case what that may be. What we do know is there is a change and the proteins are no longer associated with the organelles they were assigned in the other condition with high probability. We have included this information in the Supporting Tables for readers.

The unknown proteins i.e. the proteins which are left unclassified following the TAGM classification, are denoted by grey points in all t-SNE maps throughout the manuscript. We thank the reviewer for highlighting this and we apologise that the unknowns are missing from the legends in all the t-SNE plots. We have now updated all the figures to include the correct legend key and also referred to the unknowns explicitly in the legend captions.

The manuscript mentions “Proteins were ranked according to highest L2 distance and a stringent threshold was applied (posterior probability > 0.999; outlier probability < 1e-6)” I understand from other parts of the text that this posterior probability refers to the specific location, not to the relocalization, or does it? The authors furthermore write “events were filtered for the natural L2 distance between TAGM joint posterior probabilities.” But later only mention a ranking, not an explicit filtering at e.g. some cutoff value.

We can confirm that the posterior probability refers to the probability that a protein resides in the location it is assigned by TAGM and it is not a probability of a relocalisation. The L2 distance metric was calculated for every protein and used as a second measure on which to extract interesting movers for further data mining. The L2 distance was also used to define type 4 translocations. We have removed the above sentence and updated the main text and methods to clarify this.

Please see the main text on page 15

Based on the combination of joint posterior and outlier probabilities, protein trans/relocalisation events that occur following LPS stimulation were classified as four distinct scenarios: (i) Type 1: organelle-to-organelle, (ii) Type 2: unknown-to-organelle, (iii) Type 3: organelle-to-unknown, and (iv) Type 4: unclassified proteins that exhibit large changes in their posterior probability distribution between conditions. For additional stringency and to capture large movements in probability space, all potential relocalisation events were ranked according to their proteins natural L2 distance between their TAGM joint posterior probabilities. A large L2 distance between probabilities is indicative of proteins that exhibit large movements in probability space.

Methods on page 39

For every protein the natural L2 distance (also known as the Euclidean norm) was calculated between the TAGM joint posterior probabilities providing an extra source of information on which to rank proteins of interest. A large L2 distance implies a large change in probability distribution. The L2 distance is denoted by

$$d_{L2norm}(x, y) = \sqrt{\sum_{i=1}^n (x_i - y_i)^2}$$

where x and y are the posterior probabilities for for the unstimulated and 12h-LPS stimulated respectively, for each i th class. The L2 distance was also used for defining

potential type 4 translocations where proteins were left as “unknown” and did not meet the criteria to be assigned to one of the organelle classes in the training data. Proteins with the maximum L2 distance of 1 were extracted as type 4 for further data mining. This small subset comprised of 30 proteins.

The manuscript writes about proteins moving from one location to another, but to what extent are the proteins actually moving, or are newly produced proteins targeted to different locations (and the old ones degraded)?

Our LOPIT data, in common with the majority of dynamic proteomics studies, do not reveal anything about protein synthesis and degradation rates. If a protein is seen to move location it could be trafficking of a protein pool from a to b, or degradation of a pool of the protein in one location and synthesis and trafficking of a new pool of that protein to the alternate location. To distinguish between the two an experimental design incorporating some way of measuring newly translated proteins, e.g. pulsed SILAC or incorporation for azido-homoalanine would need to be employed. What our data do reveal is that there is a dynamic redistribution of the protein and that the protein functions at a new location upon perturbation, in this case the application of LPS.

Proteins actually leaving the mitochondria (the authors talk about “export” here) is associated with apoptosis, so I guess not all cases of proteins being in different locations after LPS are cases of actual relocalizations. Twelve hours is enough time for transcription/translation to take place after an LPS stimulus as the authors show by finding proteins with different overall concentrations (the lack of a difference for any specific protein is IMHO not enough of an argument that it actually moved)

Please see above.

Moreover, if synthesis and degradation rates are balanced, then even at a 12 hour time point we do not know whether newly synthesised proteins traffic differentially or if proteins in one specific location are degraded preferentially. Measuring the synthesis and degradation rates of proteins would be part of a much larger study.

We have modified the text accordingly to clarify these points.

Supplement Figure 2a: if the authors stick to the proteins that move between known locations (leave out the unknowns), they might be able to use arrows or so, to make a more convincing case about relocalizations.

The circo plot in Supplementary Figure 2A has directional arrows showing the flow of proteins to/from and between organelles for all 253 translocating proteins. Using arrows to annotate movement on a single t-SNE plot would be very messy, even if we only include the 112 type 1 translocations. The alluvial plot in Figure 4C in the main text also shows the flow of proteins to/from subcellular compartments for the 253 translocating proteins. We have also added alluvial plots to show the cytosolic, nuclear and lysosomal translocations depicted on the t-SNE maps in Supplementary Figures 2B-D to further clarify the flow of translocating proteins.

“Three clusters from the Bayesian temporal clustering analysis” why these three?

We decided to choose just a handful of clusters to highlight in the main text that show the information we gain from both a spatial and temporal viewpoint. We have generated a freely available online interactive app for the community that shows all annotation so the community can follow up and investigate other clusters of interest.

Where does the discrepancy between figures 4A and supplementary Figure 1G come from? The numbers are very different.

We thank the reviewer for bringing to light this error. The heatmap in Supplementary Figure 1G has been updated and the number of classifications reported has been updated on page 14 in the main text from 1,718 to 1,717 in the unstimulated, and from 1,722 to 1,713 in LPS.

We have also added a summary table of the protein markers used for TAGM classification and the number of new classifications after TAGM-MCMC to Supporting Information Tables 6 and 7 for clarity.

Regarding “Protein-protein interaction partners are also preserved during the hyperLOPIT protocol (Figure 7C). For example, we see co-localisation of (i) CDC42 and TRIP10 (CDC42-interacting protein 4)73” In 7C the proteins are close together before stimulation, but not anymore after stimulation, so we are only supposed to look at the left panel? With respect to NLRX1 and FASTKD5. I only see one dot? Are they on top of each other? If so, that might have been mentioned explicitly.

We highlight a selection of *condition specific* known protein-protein interaction partners in Figure 7C. CDC42 and TRIP10 are known to co-localise in the unstimulated state which is further shown on the unstimulated t-SNE map on the left. By looking at the right hand plot which shows the proteins after LPS stimulation, these proteins are no longer co-localised on the plot. By looking at the protein protein pairs in both conditions (the left map and the right) we see condition specific protein-protein interactions.

We thank the reviewer for noticing an error with FASTKD5. By mistake it has not been rendered in the spatial map. We have now corrected this and it is highlighted correctly.

I am not convinced by the added value of the information of supplemental figure 5. This appears rather superficial comparing of proteomics data, without a specific conclusion.

This supplementary figure was included to show the use of the spatial maps generated in this study and how they can be used as a scaffold for other studies and for proteins of interest.

Smaller things that deserve some explanation: Why would “cellular respiration” be reduced after LPS?

It has previously been demonstrated that THP-1 cell respiration is substantially altered upon LPS stimulation and is associated with a shift from oxidative phosphorylation to aerobic glycolysis (“Warburg effect”) (Ubanako et al., 2019, PLoS One, doi: 10.1371/journal.pone.0222614). LPS induces major changes in the mitochondrial electron transport chain (ETC) and Krebs’s cycle resulting in the increased generation of citrate, succinate and ROS, all of which act in a pro-inflammatory manner to fight bacterial infection (Comprehensive review on the metabolic changes in macrophages upon immune stimulation - <https://doi.org/10.1002/eji.201445427>). Bayesian Temporal Clustering

allocated seven mitochondrial proteins (NDUFV3, COX4I1, ETFA, NDUF57, SUCLG1, FH and DHD) associated with ETC and Krebs's cycle to cluster 11, which reflects the metabolic changes occurring upon LPS stimulation.

We have not included this paragraph in the manuscript as this represents only a small cluster from the temporal clustering analysis and the paper cannot accommodate any more text due to size limitations. We feel there are many interesting biological features in this dataset and it is unfortunate that we cannot discuss all of them within the constraints of a manuscript.

*and "CD14, the classic co-receptor for LPS, was localised to the PM in unstimulated conditions, however was no longer associated with the PM following stimulation"
so where did it go?*

CD14 was classified to the plasma membrane (PM) in the unstimulated condition and was unclassified in the LPS-stimulated condition. As mentioned above, this suggests that CD14 may be decoupled from the PM in response to LPS. There is a very strong literature base to support this suggestion, as the movement of CD14 from the PM to the endosomal compartment in coordination with TLR4 upon LPS stimulation has clearly been shown. (Cell. 2011 Nov 11;147(4):868-80. doi: 10.1016/j.cell.2011.09.051.) In addition, we have indirect evidence to support a relocalisation of the CD14/TLR4 complex to the endosomal compartment in our experiments. The time course results (Figure 2B,C) identifies an elevation in a number of proteins (MX1, MX2 and IFIT3) that are known to be dependent on TLR4-TRIF signalling (Eur J Immunol. 2004 Feb;34(2):558-64. doi: 10.1002/eji.200324714.) and a prerequisite for TLR4-TRIF signalling is the CD14 assisted relocalisation of PM associated TLR4 to the endosomal compartment (Cell. 2011 Nov 11;147(4):868-80. doi: 10.1016/j.cell.2011.09.051.) We can therefore speculate that CD14 localises to an endosomal compartment upon LPS stimulation and because endosomes were not well characterised in the LOPIT analysis, CD14 localisation remained unclassified in our analysis.

It is important to note that CD14 was not identified as a mover in this analysis and the text has been updated to clarify this.

We hope that our revisions and corrections are to the satisfaction of the reviewers.

Yours sincerely,

Kathryn Lilley

Professor of Cellular Dynamics
Director of the Cambridge Centre for Proteomics
Department of Biochemistry and Milner Therapeutics Institute
Fellow of Jesus College, Cambridge

REVIEWERS' COMMENTS

Reviewer #1 (Remarks to the Author):

The manuscript has been significantly improved by the authors. I am satisfied with the revision and suggest this work to be published.

Reviewer #3 (Remarks to the Author):

I have no more issues with this manuscript. Nice work!